# Reinforcement Learning in a New Keynesian Model [†]

Szabolcs Deák [1], Paul Levine [2,*], Joseph Pearlman [3] and Bo Yang [4]

1    Department of Economics, University of Exeter, Exeter EX4 4PU, UK; s.deak@exeter.ac.uk
2    School of Economics, University of Surrey, Guildford GU2 7XH, UK
3    Department of Economics, City University London, London EC1R 0JD, UK; joseph.pearlman.1@city.ac.uk
4    Department of Economics, Swansea University, Swansea SA2 8PP, UK; bo.yang@swansea.ac.uk
*    Correspondence: p.levine@surrey.ac.uk
†    This paper is an extended version of our paper presented at the CEF 2015 Conference, Taiwan, June 2015; the MMF 2016 Conference, University of Bath, September 2016; workshops at the Bank of England, March 2015, the Tinbergen Institute, October, 2015 and the Bank of Portugal, June 2018.

**Abstract:** We construct a New Keynesian (NK) behavioural macroeconomic model with bounded-rationality (BR) and heterogeneous agents. We solve and simulate the model using a third-order approximation for a given policy and evaluate its properties using this solution. The model is inhabited by fully rational (RE) and BR agents. The latter are anticipated utility learners, given their beliefs of aggregate states, and they use simple heuristic rules to forecast aggregate variables exogenous to their micro-environment. In the most general form of the model, RE and BR agents learn from their forecasting errors by observing and comparing them with each other, making the composition of the two types endogenous. This reinforcement learning is then at the core of the heterogeneous expectations model and leads to the striking result that increasing the volatility of exogenous shocks, by assisting the learning process, increases the proportion of RE agents and is welfare-increasing.

**Keywords:** new Keynesian behavioural model; heterogeneous expectations; bounded rationality; reinforcement learning

## 1. Introduction

Since the burst of the United States housing bubble in 2008, a large amount of recent behavioural macroeconomics literature has emerged in response to what many regard as the extreme modelling assumption of rational (model-consistent) expectations—henceforth RE. Its defining characteristic is to limit the cognitive skills of at least a group of agents in the model. One strand of this literature achieves this by introducing simple 'heuristic' learning rules which can be thought of as parsimonious forms of forecasting rules (as in References [1–4]). This, we argue, fits well the behavioural approach of assuming agents in the model with limited cognitive skills who behave according to bounded rationality—henceforth BR.

However, this raises the opposite concern regarding the bounds on BR: with heuristic rule-of-thumb behaviour, agents may fall considerably short of building RE, and such models are particularly vulnerable to the Lucas critique when policy scenarios are studied. The problem is that agents can depart from rationality in an infinite number of ways, leading into the "the wilderness of bounded rationality problem" of Reference [5]. The challenge posed by the wilderness is clearly demonstrated by the sheer size of literature on behavioural macroeconomics and the huge number of equilibria proposed. Surveys include References [6–9].

The concern of behavioural models regarding RE are shared by the recent Agent-Based(AB) alternatives. This approach represents economic agents as well as various social and environmental phenomena as autonomous virtual entities that interact during simulation experiments following pre-defined rules. In standard macroeconomic models, agents'

decisions consist of behavioural equations or, in the case of dynamic stochastic general equilibrium (DSGE) models, micro-founded first-order conditions satisfying a dynamic optimisation problem, that are continuous functions of the current and past state of the economy. The AB approach provides a potentially more flexible way of modelling the cognitive capabilities of decision makers and their responses to both the macro- and individual micro-environment (for example, the authors of Reference [10] studied the inter-linkages between the real and financial sides of the economy using an AB framework in which different types of agents interact on different markets following simple heuristic rules).

When emotional states, cognitive limitations and past information play a key role in economic behaviour, the AB decision process serves as a promising approach for accounting for the behaviour of heterogeneous rule-possessing agents. In AB models, economies can represent out-of-equilibrium behaviour and non-market clearing and can be regarded as "evolving systems of autonomous interacting agents" Reference [11]. Hence, while DSGE assumes that agents have very sophisticated computational capabilities and live in very simple environments, AB models assume that people use simple behavioural rules to cope with complex and dynamic environments. Many of the features of AB models in addition to non-RE, such as heterogeneous agents and unemployment, are now being incorporated into DSGE models. The bounded-rational behavioural models with learning can be then seen as a genre with both classical DSGE and AB modelling features (see Reference [12] for further discussions).

In response to the wilderness concern, the literature on BR models adopts a basic general heterogeneous expectations framework pioneered by Reference [13]. To limit the departure from rationality, the approach of *reinforcement learning* proposes that, although adaptation can be slow and there can be a random component of choice, the higher the "payoff" (defined appropriately) from taking an action in the past, the more likely it will be taken in the future. We adopt a heterogeneous RE-BR model of this type. The idea behind this correction mechanism in which agents evaluate the payoff function is rooted in discrete choice theory, which is extensively studied in the fields of experimental economics and cognitive psychology. Recent studies have shown that, when managing their incentive structures, agents with market-consistent information may not follow rational choice theory and do not always correct irrational behaviour even if they have sufficient knowledge available to correct it Reference [14]. Instead, a recent study by Reference [15] conducted several experiments to analyse how agents decide between different alternatives. The results showed that people tend to evaluate their perceived efficacy to correct the error by following rational principles based on cognitively assessing the costs and benefits (payoff) associated with the correction.

In addition to the selection mechanism, for given proportions of RE and BR agents, there then exists a choice of learning model: *Euler* versus the *anticipated utility* approach (following Reference [16])—henceforth EL and AU. In both approaches, agents cannot form model-consistent expectations. Under EL, agents forecast their own one-period-ahead decisions, whereas under AU, agents form beliefs over the future infinite time horizon of aggregate states and prices which are exogenous to their decisions (AU, also known the "infinite time-horizon" framework, is closely related to the "internal rationality" (IR) approach of Reference [17]). Under both IR and AU, agents maximise utility, given their constraints and a consistent set of probability beliefs about payoff-relevant variables that are external. Then with IR, beliefs take the form of a well-defined probability measure over a stochastic process (the "fully Bayesian" plan). The authors of Reference [18] compared the IR vs. AU and found that AU can closely approximate the fully Bayesian optimisation. The two approaches then differ with respect to what agents learn about—their own future one-period ahead decision for EL and variables exogenous to the agents for AU.

In this paper, we introduce heterogeneity in a full Brock–Hommes new Keynesian (NK) model with a composite specification of BR and RE agents allowing for a wealth distribution between the two groups. A third-order perturbation solution leads to a demonstration of the effects of reinforcement learning in our NK boundedly rational model environment.

The primary interest of this paper is to study the effect of learning on the business cycle and its implications for the design of optimal policy strategies within the BR environment. To this end, the discussions are organised around a number of issues that we aim to address. Can our model with an endogenous selection mechanism generate endogenous persistence and non-normality in the frequency distribution of macroeconomic aggregates? Does the composition of the types of agents change with reinforcement learning and the nature of the shocks hitting the economy? What are the welfare implications based on a behavioural macroeconomic model of this type?

In particular, the main contributions of this paper are as follows: (1) we develop a micro-founded framework that models the endogenous composition of RE and non-RE agents with reinforcement learning along the lines of Reference [19]; (2) we carry out our simulations based on different parameterisations of the model and focus on an assessment of the model-implied moments, including the simulated impulse response functions. Furthermore, in Appendices A–G we discuss the sources of instability and indeterminacy in our setup featuring the BR agents who solve their decision problems using the EL and AU expectation formation schemes. The highly non-linear structure of the BR specification in which agents endogenously select the heuristic rules is crucial for conducting optimal policy in macroeconomic models.

Our paper aims to contribute to both the learning and macroeconomic literature. The investigation on the role of BR behaviour in understanding the dynamics in economic activity observed empirically and guiding policy choices is not a trivial one. Various attempts modify the baseline NK model to account for hybrid heterogeneous expectations and BR. An approach that is closely related to ours in this regard is from the earlier contributions of References [3,19,20], in which they studied calibrated composite heterogeneous expectations models of RE and BR agents and discuss implications for the business cycle and designing stabilisation policies. In our setting, we focus on the major BR approaches with reinforcement learning—a highly non-linear structure within BR which is methodologically relevant for capturing movements that are non-normally distributed in empirical data. We also investigate the effect on rationality when we subject our model to the occurrence of more volatile exogenous shocks.

The rest of the paper is structured as follows. Section 2 sets out the standard linear RE NK model used in the literature and then proceeds to the Brock–Hommes composite model of rational and boundedly rational agents. Section 3 goes back to the non-linear foundations of the model. Section 4 describes the specific market-consistent environment in which households and firms form their expectations. Then, Section 5 presents our main results. Section 5.3 discusses how we choose the set of parameter values that avoids chaotic dynamics. Finally, Section 6 concludes the paper. Appendices A–G contain further details and results on the model's stability and the construction of the model.

## 2. The Standard Behavioural NK Model

This section discusses the standard behavioural NK model framework used by References [3,4,19–22] and others.

### 2.1. The Workhorse NK Model

We first set out the most basic three-equation linearised workhorse NK model with RE

$$y_t = \mathbb{E}_t y_{t+1} - (r_{n,t} - \mathbb{E}_t \pi_{t+1}) + u_{1,t} \tag{1}$$

$$\pi_t = \beta \mathbb{E}_t \pi_{t+1} + \kappa y_t + u_{2,t} \tag{2}$$

$$r_{n,t} = \rho_r r_{n,t-1} + (1 - \rho_r)(\theta_\pi \pi_t + \theta_y y_t) + u_{3,t} \tag{3}$$

where $y_t$, $\pi_t$ and $r_{n,t}$ are the output gap, the inflation rate and the nominal interest rate, respectively. All variables are expressed in log-deviation form about a zero net-inflation steady state. The shock processes $u_{i,t}$, $i = 1, 2, 3$ should be interpreted as exogenous shocks to demand (or preferences), the supply side, and monetary policy, respectively, and they

are usually AR(1) processes. Expectations ($\mathbb{E}_t$) up to now are formed, assuming RE and perfect information of the state vector (which includes the shock processes). Equation (1) is the linearised Euler equation for consumption which is equated with output in equilibrium (there is no government expenditure). The value (2) is the NK Phillips curve, and (3) is the nominal interest rate rule in "implementable form" in that it responds to output relative to the steady state rather than the output gap (note that (1) assumes logarithmic utility and that the supply side shock is a composite of technology and marginal cost processes in the model developed in this paper. The AR(1) feature of shock processes is criticised by Reference [4], as it implies that persistence is exogenously generated. This paper addresses this critique in developing strong endogenous persistence mechanisms through learning).

Before relaxing the RE assumption, two points about this formulation need to be made. First, there are not a lagged term in $y_t$ in the demand curve (1) nor a lagged term in $\pi_t$ in the Phillips curve (2) (as, for example, in Reference [23]). These can enter through the introduction of external habits in the consumers' utility function and price indexing, respectively, but we choose to focus on learning as a persistence mechanism; thus, both these features are omitted. Second, the linearisation even without these persistence terms is only correct for a zero-inflation steady state.

### 2.2. The Brock–Hommes Behavioural NK Model

In the Brock–Hommes framework, which we later follow, the model becomes behavioural by a departure from the RE assumption and the introduction of two groups of agents. One group is rational, and the other forms EL expectations through simple "heuristic" learning rules. RE agents form model-consistent expectations fully aware of the existence of BR agents in the composite model. A version of general adaptive learning rules (the authors of Reference [24] provided lab-based support for such rules, and the generalised heuristic rule we later adopt in Section 4 includes a $t - 2$ period and encompasses all the different behavioural group forecast heuristics) that encompasses those adopted by References [3,4,13,19,25] is

$$\mathbb{E}_t^* y_{t+1} = \mathbb{E}_{t-1}^* y_t + \lambda_y (y_{t-j} - \mathbb{E}_{t-1}^* y_t); \quad \lambda_y \in [0,1], \, j = 0,1 \tag{4}$$

$$\mathbb{E}_t^* \pi_{t+1} = \mathbb{E}_{t-1}^* \pi_t + \lambda_\pi (\pi_{t-j} - \mathbb{E}_{t-1}^* \pi_t); \quad \lambda_\pi \in [0,1], \, j = 0,1 \tag{5}$$

where we can in principle allow for both current and lagged observations of output and inflation, $j = 0, 1$, respectively. Throughout the rest of the paper, we make the following information assumptions: for observations of aggregateoutput and inflation, similar to the EL approach, we assume $j = 1$. Later in the AU approach, we need to model observations of market-specificvariables consisting of factor prices, profits and marginal costs. These we assume can be observed without a lag, and therefore, $j = 0$.

Let $n_{y,t}$, $n_{\pi,t}$ be the proportions of rational agents forecasting output and inflation, respectively. The IS and NK Phillips curve equations then become

$$y_t = n_{y,t} \mathbb{E}_t y_{t+1} + (1 - n_{y,t}) \mathbb{E}_t^* y_{t+1} - [r_{n,t} - (n_{\pi,t} \mathbb{E}_t \pi_{t+1} + (1 - n_{\pi,t}) \mathbb{E}_t^* \pi_{t+1})] + u_{1,t} \tag{6}$$

$$\pi_t = \beta[n_{\pi,t} \mathbb{E}_t \pi_{t+1} + (1 - n_{\pi,t}) \mathbb{E}_t^* \pi_{t+1}] + \lambda y_t + u_{2,t} \tag{7}$$

To complete the model, we need expressions for the weights $n_{y,t}$ and $n_{\pi,t}$. These follow the reinforcement learning literature by choosing probabilities

$$n_{x,t} = \frac{\exp(-\gamma \Phi_{x,t}^{RE}(\{x_t\}))}{\exp(-\gamma \Phi_{x,t}^{RE}(\{x_t\})) + \exp(-\gamma \Phi_{x,t}^{AE}(\{x_t\}))} \tag{8}$$

where $-\Phi_{x,t}^{RE}(\{x_t\})$ and $-\Phi_{x,t}^{AE}(\{x_t\})$ are "fitness" measures, respectively, of the forecast performance of the rational and non-rational predictor of outcome $\{x_t\} = \{y_t\}, \{\pi_t\}$ given by a discounted least-squares error predictor

$$\Phi_{x,t}^{RE}(\{x_t\}) = \mu_{RE}\Phi_{x,t-1}^{RE}(\{x_t\}) + (1-\mu_{RE})([x_t - \mathbb{E}_{t-1}x_t]^2 + C_x) \qquad (9)$$

$$\Phi_{x,t}^{AE}(\{x_t\}) = \mu_{AE}\Phi_{x,t-1}^{AE}(\{x_t\}) + (1-\mu_{AE})[x_{t-j} - \mathbb{E}_{t-1-j}^* x_{t-1}]^2 ; j = 0, 1 \qquad (10)$$

where $\mu_{RE}$ and $\mu_{AE}$ capture the memory of the agents forming RE and adaptive expectations (a measure of forgetfulness of past observations). $C_x$ represents the relative costs of being rational in learning about variable $x_t$. Thus, the proportion of rational agents in the steady state is given by

$$n_x = \frac{\exp(-\gamma C_x)}{\exp(-\gamma C_x) + 1}$$

which is pinned down by the $\gamma C_x$. Equations (3)–(10) constitute the linearised NK behavioural model (the authors of References [3,4] constructed a rather different composite EL-type model consisting of "fundamentalist" rather than rational agents alongside adaptive learners. For the former RE, $\mathbb{E}(\cdot)$ are replaced with $\mathbb{E}^f y_{t+1} = y_t^F$ and $\mathbb{E}^f \pi_{t+1} = 0$. Thus, fundamentalists always believe that the next period's output gap is zero and that the net inflation rate will return to its steady-state value of zero. The same authors also assume $C_x = 0$ in (9)).

## 3. The Non-Linear NK Model

Thus far in the linearised model, the justification for the form of adaptive forecasts needs to be established. In order to address this, we step back to the underlying non-linear model and introduce the distinction between internal decisions and aggregate macrovariables. We start with the non-linear RE model and proceed from full to bounded rationality in stages. The complete model setup and its balanced growth steady state are summarised in Appendices A–G.

### 3.1. Households

Household $j$ chooses savings between work and labour supply. Let $C_t(j)$ be consumption and $H_t(j)$ be the proportion of available work or leisure spent at the former. The single-period utility we choose, compatible with a balanced growth steady state, is

$$U_t(j) = U(C_t(j), H_t(j)) = \log(C_t(j)) - \frac{H_t(j)^{1+\phi}}{1+\phi}$$

and the value function of the representative household at time $t$ dependent on its assets $B$ is

$$V_t(j) = V_t(B_{t-1}(j)) = \mathbb{E}_t\left[\sum_{s=0}^{\infty} \beta^s U(C_{t+s}(j), H_{t+s}(j))\right] \qquad (11)$$

The household's problem at time $t$ is to choose paths for consumption $\{C_t(j)\}$, labour supply $\{H_t(j)\}$ and holdings of financial savings to maximise $V_t(j)$, given by (11), given its budget constraint in period $t$

$$B_t(j) = R_t B_{t-1}(j) + W_t H_t(j) + \Gamma_t - C_t(j) - T_t - \frac{\omega}{2}(B_{t-1}(j) - B)^2 \qquad (12)$$

where $B_t(j)$ is the given net stock of real financial assets at the end of period $t$, $W_t$ is the wage rate, $T_t$ are lump-sum taxes, and $\Gamma_t$ are profits from wholesale and retail firms owned by households. In order to allow for a wealth distribution by heterogenous agents introduced later and to achieve a stationary path for bond holdings, we introduce a portfolio adjustment cost (this as a modelling device similar to that used in open economies with home and foreign household is pioneered by Reference [26]. We examine the limit as $\omega$ becomes

very small so that our choice of real rather than nominal bond holding costs is immaterial. The wealth distribution effect does not significantly change the equilibrium). $R_t$ is the real interest rate paid on assets held at the beginning of period $t$ given by $R_t = \frac{R_{n,t-1}}{\Pi_t} RS_t$, where $R_{n,t}$ and $\Pi_t$ are the nominal interest and inflation rates, respectively, and $RS_t$ is a risk premium shock. $W_t$, $R_{n,t}$, $\Pi_t$ and $\Gamma_t$ are all exogenous to household $j$. As usual, all real variables are expressed relative to the price of the final output. The standard first-order conditions are

$$
\begin{aligned}
\mathbb{E}_t[\Lambda_{t,t+1}(j) R_{t+1}] &= 1 + \varpi(B_t(j) - B) \\
\frac{U_{H,t}(j)}{U_{C,t}(j)} &= -W_t
\end{aligned}
$$

where $\Lambda_{t,t+1}(j) \equiv \beta \frac{U_{C,t+1}(j)}{U_{C,t}(j)}$ is the stochastic discount factor for household $j$, over the interval $[t, t+1]$. For our choice of utility function $U_{C,t} = \frac{1}{C_t}$ and $U_{H,t} = -H_t^\phi$, and these become

$$
\beta \mathbb{E}_t \left[ \frac{C_t(j) R_{t+1}}{C_{t+1}(j)} \right] = 1 + \varpi(B_t(j) - B) \tag{13}
$$

$$
C_t(j) H_t(j)^\phi = W_t \Rightarrow H_t(j) = \left( \frac{W_t}{C_t(j)} \right)^{\frac{1}{\phi}} \tag{14}
$$

The first-order conditions up to now are suitable for the RE solution. We now express the solution in a form suitable for moving from an RE to a learning equilibrium. We consider the limit as $\varpi \to 0$. Solving (12) forward in time and imposing the transversality condition on debt, we can write

$$
B_{t-1}(j) = \mathrm{PV}_t(C_t(j)) - \mathrm{PV}_t(W_t H_t(j)) - \mathrm{PV}_t(\Gamma_t) + \mathrm{PV}_t(T_t) \tag{15}
$$

where the present (expected) value of a series $X \equiv \{X_{t+i}\}_{i=0}^{\infty}$ at time $t$ is defined by

$$
\mathrm{PV}_t(X_t) \equiv \mathbb{E}_t \sum_{i=0}^{\infty} \frac{X_{t+i}}{R_{t,t+i}} = \frac{X_t}{R_t} + \frac{1}{R_t} \mathrm{PV}_t(X_{t+1}) \tag{16}
$$

writing $R_{t,t+i} \equiv R_t R_{t+1} R_{t+2} \cdots R_{t+i}$ as the real interest rate over the interval $[t-1, t+i]$.

The forward-looking budget constraint (15) holds for the representative household. If we allow RE and BR agents to borrow from or lend to one another, we must allow for $B_{t-1} \neq 0$. Then, in a symmetric equilibrium with $C_t(j) = C_t$ and $H_t(j) = H_t$, (15) and (14) become

$$
\begin{aligned}
B_{t-1} &= \mathrm{PV}_t(C_t) - \mathrm{PV}_t \left( \frac{W_t^{1+\frac{1}{\phi}}}{C_t^{\frac{1}{\phi}}} \right) - \mathrm{PV}_t(\Gamma_t) + \mathrm{PV}_t(T_t) \\
H_t &= \left( \frac{W_t}{C_t} \right)^{\frac{1}{\phi}}
\end{aligned}
$$

Solving (13) forward in time and using the law of iterated expectation, we have for $i \geq 1$

$$
\frac{1}{C_t} = \beta^i \mathbb{E}_t \left[ \frac{R_{t+1,t+i}}{C_{t+i}} \right]; \ i \geq 1 \tag{17}
$$

We now express the solution to the household optimisation problem for $C_t$ and $H_t$ that are functions of *point expectations* $\{\mathbb{E}_t W_{t+i}\}_{i=1}^{\infty}$, $\{\mathbb{E}_t R_{t+1,t+i}\}_{i=1}^{\infty}$ and $\{\mathbb{E}_t \Gamma_{t+i}\}_{i=0}^{\infty}$, treated as

exogenous processes given at time $t$. With point expectations, we use (17) to obtain the following optimal decision for $C_{t+i}$, given the point expectations $\mathbb{E}_t R_{t+1,t+i}$

$$C_{t+i} \;=\; C_t \beta^i \mathbb{E}_t R_{t+1,t+i} \;;\; i \geq 1 \tag{18}$$

$$\mathbb{E}_t(W_{t+i} H_{t+i}) \;=\; \frac{\left(\mathbb{E}_t W_{t+i}\right)^{1+\frac{1}{\phi}}}{C_{t+i}^{\frac{1}{\phi}}} \tag{19}$$

Substituting (18) and (19) into the forward-looking household budget constraint, using $\sum_{i=0}^{\infty} \beta^i = \frac{1}{1-\beta}$ and $\mathbb{E}_t R_{t,t+i} = R_t \mathbb{E}_t R_{t+1,t+i}$ for $i \geq 1$, we arrive at

$$\frac{C_t - R_t B_{t-1}}{(1-\beta)} = \frac{1}{C_t^{\frac{1}{\phi}}} \left( W_t^{1+\frac{1}{\phi}} + \sum_{i=1}^{\infty} (\beta^{\frac{1}{\phi}})^{-i} \left( \frac{\mathbb{E}_t W_{t+i}}{\mathbb{E}_t R_{t+1,t+i}} \right)^{1+\frac{1}{\phi}} \right) + \Gamma_t - T_t + \sum_{i=1}^{\infty} \frac{\mathbb{E}_t(\Gamma_{t+i} - T_{t+i})}{\mathbb{E}_t R_{t+1,t+i}}$$

which can be written in recursive form as

$$\frac{C_t - R_t B_{t-1}}{(1-\beta)} = \frac{1}{C_t^{\frac{1}{\phi}}} \left( W_t^{1+\frac{1}{\phi}} + \Omega_{1,t} \right) + \Gamma_t - T_t + \Omega_{2,t} \tag{20}$$

$$\Omega_{1,t} \equiv \sum_{i=1}^{\infty} (\beta^{\frac{1}{\phi}})^{-i} \left( \frac{\mathbb{E}_t W_{t+i}}{\mathbb{E}_t R_{t+1,t+i}} \right)^{1+\frac{1}{\phi}} = (\beta^{\frac{1}{\phi}})^{-1} \left( \frac{\mathbb{E}_t W_{t+1}}{\mathbb{E}_t R_{t+1,t+1}} \right)^{1+\frac{1}{\phi}} + \frac{\Omega_{1,t+1}}{\beta^{\frac{1}{\phi}} \mathbb{E}_t R_{t+1}}$$

$$\Omega_{2,t} \equiv \sum_{i=1}^{\infty} \frac{\mathbb{E}_t(\Gamma_{t+i} - T_{t+i})}{\mathbb{E}_t R_{t+1,t+i}} = \frac{\mathbb{E}_t(\Gamma_{t+1} - T_{t+1})}{\mathbb{E}_t R_{t+1,t+1}} + \frac{\Omega_{2,t+1}}{\mathbb{E}_t R_{t+1}}$$

Consumption is then given by (20), assuming point expectations or by the symmetric form of the Euler equation (13) under full rationality (i.e., households know the symmetric nature of equilibrium with $C_t(j) = C_t$). $C_t$ is a function of *rational point expectations* $\{\mathbb{E}_t W_{t+i}\}_{i=1}^{\infty}$, $\{\mathbb{E}_t R_{t,t+i}\}_{i=i}^{\infty}$ and $\{\mathbb{E}_t \Gamma_{t+i}\}_{i=1}^{\infty}$ which can be treated as exogenous processes given at time $t$ or as rational model-consistent expectations. Since $E_t f(X_t) \approx f(E_t(X_t))$; $E_t f(X_t Y_t)) \approx f(E_t(X_t) E_t(Y_t))$ up to a first-order Taylor-series expansion, assuming that point expectations are equivalent to using a linear approximation (given below), as is usually performed in the literature.

### 3.2. Firms, Government Expenditures and Monetary Policy

This section sets out the wholesalers and the retail sector which is optimised using Calvo-pricing contracts. We close the non-linear setup with resource and balanced government budget constraints, a monetary policy rule and by specifying the structural shocks in the economy. Wholesale firms employ a Cobb–Douglas production function to produce a homogeneous output

$$Y_t^W = F(A_t, H_t) = A_t H_t^{\alpha}$$

where $A_t$ is total factor productivity. Profit-maximising demand for labour results in the first-order condition

$$W_t = \frac{P_t^W}{P_t} F_{H,t} = \alpha \frac{P_t^W}{P_t} \frac{Y_t^W}{H_t} \tag{21}$$

The retail sector costlessly converts a homogeneous wholesale good into a basket of differentiated goods for aggregate consumption

$$C_t = \left( \int_0^1 C_t(m)^{(\zeta-1)/\zeta} dm \right)^{\zeta/(\zeta-1)} \tag{22}$$

where $\zeta$ is the elasticity of substitution. For each $m$, the consumer chooses $C_t(m)$ at a price $P_t(m)$ to maximise (22) given total expenditure $\int_0^1 P_t(m) C_t(m) dm$. Assuming that

government services are similarly differentiated, this results in a set of demand equations for each differentiated good $m$ with price $P_t(m)$ of the form

$$Y_t(m) = \left(\frac{P_t(m)}{P_t}\right)^{-\zeta} Y_t \tag{23}$$

where $P_t = \left[\int_0^1 P_t(m)^{1-\zeta} dm\right]^{\frac{1}{1-\zeta}}$, $P_t$ is the aggregate price index, and $C_t$ and $P_t$ are Dixit–Stigliz aggregates; see Reference [27].

Following Reference [28], we assume that there is a probability of $1 - \xi$ at each period that the price of each retail good $m$ is set optimally to $P_t^O(m)$. If the price is not re-optimised, then it is held fixed. For each retail producer $m$, given its real marginal cost $MC_t = \frac{P_t^W}{P_t}$, the objective is at time $t$ to choose $\{P_t^O(m)\}$ to maximise discounted real profits

$$\mathbb{E}_t \sum_{k=0}^{\infty} \xi^k \frac{\Lambda_{t,t+k}}{P_{t+k}} Y_{t+k}(m)\left[P_t^O(m) - P_{t+k}MC_{t+k}\right]$$

subject to (23), where $\Lambda_{t,t+k} \equiv \beta^k \frac{U_{C,t+k}}{U_{C,t}}$ is the stochastic discount factor over the interval $[t, t+k]$. The solution to this is standard and is given by

$$\frac{P_t^O(m)}{P_t} = \frac{\zeta}{\zeta - 1} \frac{\mathbb{E}_t \sum_{k=0}^{\infty} \xi^k \Lambda_{t,t+k}(\Pi_{t,t+k})^{\zeta} Y_{t+k} MC_{t+k}}{\mathbb{E}_t \sum_{k=0}^{\infty} \xi^k \Lambda_{t,t+k}(\Pi_{t,t+k})^{\zeta}(\Pi_{t,t+k})^{-1} Y_{t+k}}$$

Denoting the numerator and denominator by $J_t$ and $JJ_t$, respectively, and introducing a mark-up shock $MS_t$ to $MC_t$, from Appendix D, we write in recursive form

$$\frac{P_t^O(m)}{P_t} = \frac{J_t}{JJ_t} \tag{24}$$

$$J_t - \xi \mathbb{E}_t[\Lambda_{t,t+1}\Pi_{t+1}^{\zeta}J_{t+1}] = \frac{1}{1 - \frac{1}{\zeta}} Y_t MC_t MS_t \tag{25}$$

$$JJ_t - \xi \mathbb{E}_t[\Lambda_{t,t+1}\Pi_{t+1}^{\zeta-1}JJ_{t+1}] = Y_t \tag{26}$$

Using the fact that all resetting firms will choose the same price, by the law of large numbers, we can find the evolution of inflation given by

$$1 = \xi(\Pi_{t-1,t})^{\zeta-1} + (1 - \xi)\left(\frac{P_t^O}{P_t}\right)^{1-\zeta} \tag{27}$$

Price dispersion lowers aggregate output as follows. Market clearing in the labour market gives

$$H_t = \sum_{m=1}^{n} H_t(m) = \sum_{m=1}^{n}\left(\frac{Y_t(m)}{A_t}\right)^{\frac{1}{\alpha}} = \left(\frac{Y_t}{A_t}\right)^{\frac{1}{\alpha}} \sum_{m=1}^{n}\left(\frac{P_t(m)}{P_t}\right)^{-\frac{\zeta}{\alpha}}$$

using (23). Hence, equilibrium for good $m$ gives $Y_t = \frac{Y_t^W}{\Delta_t^{\alpha}}$, where price dispersion is defined by

$$\Delta_t \equiv \left(\sum_{m=1}^{n}\left(\frac{P_t(m)}{P_t}\right)^{-\frac{\zeta}{\alpha}}\right)$$

Assuming that the number of firms is large from Appendix E, we obtain the following dynamic relationship

$$\Delta_t = \xi \Pi_t^{\frac{\zeta}{\alpha}} \Delta_{t-1} + (1 - \xi) \left( \frac{J_t}{JJ_t} \right)^{-\frac{\zeta}{\alpha}} \tag{28}$$

To close the model, we first require total profits from retail, and wholesale firms, $\Gamma_t$, is remitted to households. This is given in real terms by

$$\Gamma_t = \underbrace{Y_t - \frac{P_t^W}{P_t} Y_t^W}_{\text{retail}} + \underbrace{\frac{P_t^W}{P_t} Y_t^W - W_t H_t}_{\text{Wholesale}} = Y_t - \alpha \frac{P_t^W}{P_t} Y_t^W$$

using the first-order condition (21). Then, to complete closure, we have resource and balanced government budget constraints

$$Y_t = C_t + G_t = C_t + T_t$$

where $G_t$ is an exogenous demand process, and a monetary policy rule for the nominal interest rate given by the following implementable Taylor-type rule

$$
\begin{aligned}
\log \left( \frac{R_{n,t}}{R_n} \right) &= \rho_r \log \left( \frac{R_{n,t-1}}{R_n} \right) + (1 - \rho_r) \Big( \theta_\pi \log \left( \frac{\Pi_t}{\Pi_{targ,t}} \right) \\
&+ \theta_y \log \left( \frac{Y_t}{Y} \right) + \theta_{dy} \log \left( \frac{Y_t}{Y_{t-1}} \right) \Big) + \epsilon_{MP,t}
\end{aligned}
$$

and $\epsilon_{MP,t}$ is an i.i.d. shock to monetary policy. $\Pi_{targ,t}$ is a time-varying inflation target and together with $A_t$, $G_t$, $RS_t$ and $MS_t$ follows an AR(1) process. This completes the model.

### 3.3. Recovering the NK Workhorse Model

We now show that the linearised form of the non-linear model about the steady state reduces to the standard workhorse model in Section 2.1 where rational expectations $\mathbb{E}_t y_{t+1}$ and $\mathbb{E}_t \pi_{t+1}$ or non-RE $\mathbb{E}_t^* y_{t+1}$ and $\mathbb{E}_t^* \pi_{t+1}$ can be treated as expectations by individual households and firms, respectively, of *aggregate* future output and inflation. We consider the linearised form of the above set-up about a zero inflation and growth deterministic steady state. We also ignore lending or borrowing between RE and BR agents. With RE, the household $j$'s first-order conditions take one of two forms. First, linearising (20), we have

$$
\begin{aligned}
\alpha_1 c_t(j) &= \alpha_2 w_t + \alpha_3 (\omega_{2,t} + r_t) + \alpha_4 \omega_{1,t} \tag{29} \\
\omega_{1,t} &= \alpha_5 \mathbb{E}_t w_{t+1} - \alpha_6 \mathbb{E}_t r_{t+1} + \beta \mathbb{E}_t \omega_{1,t+1} \\
\omega_{2,t} &= (1 - \beta)(\gamma_t - g_t) - r_t + \beta \mathbb{E}_t \omega_{2,t+1} \\
\gamma_t &= \frac{1}{\gamma_y} y_t - \frac{\alpha}{\gamma_y} (w_t + h_t)
\end{aligned}
$$

where lower case variables $x_t \equiv \log(X_t/X)$, $X$ is the steady state of $X_t$; $c_y \equiv \frac{C}{Y}$, $\gamma_y \equiv \frac{\Gamma}{Y}$, $g_y \equiv \frac{G}{Y}$ and $\gamma_t$ is *exogenous* profit per household (a function of aggregate consumption and hours). Positive coefficients are given by $\alpha_1 \equiv 1 + \frac{\alpha}{\phi c_y}$, $\alpha_2 \equiv (1 - \beta)(1 + \frac{1}{\phi}) \frac{\alpha}{c_y}$, $\alpha_3 \equiv \frac{\gamma_y}{c_y}$, $\alpha_4 \equiv \frac{\beta \alpha}{c_y}$, $\alpha_5 \equiv (1 - \beta)(1 + \frac{1}{\phi})$ and $\alpha_6 \equiv (1 + \frac{1}{\phi})$. Alternatively, from Euler Equation (13),

$$c_t = \mathbb{E}_t c_{t+1} - \mathbb{E}_t r_{t+1} \tag{30}$$

in a symmetric equilibrium. Under RE, (29) or (30) lead to the same equilibrium, but under BR, this is no longer the case.

Linearising the household supply of hours decision, the resource constraint and the Fisher equation, we have

$$y_t = (1 - g_y)c_t + g_y g_t \tag{31}$$

$$r_t = r_{n,t-1} - \pi_t + rs_{t-1} \tag{32}$$

$$h_t = \frac{1}{\phi}(w_t - c_t)$$

Then, in a special case where $G_t = 0$ and there is no distinction between public and private consumption, $g_y = 0$ and $y_t = c_t$. Equations (30)–(32) with $rs_t = u_{1,t}$ reduce to (1) where $\mathbb{E}_t y_{t+1}$ is the forecast of *aggregate* output.

Turning to the supply side, for the wholesale sector

$$y_t = a_t + \alpha h_t$$

$$mc_t = w_t - y_t + h_t$$

For retail firm $m$, linearising the pricing dynamics (24)–(26) about a zero net equation steady state and solving forward, we have

$$
\begin{aligned}
p_t^o(m) - p_t &= \beta\xi\mathbb{E}_t[\pi_{t+1} + p_{t+1}^o(m) - p_{t+1}] + (1 - \beta\xi)(mc_t + ms_t) \\
&= \mathbb{E}_t \sum_{i=0}^{\infty} (\beta\xi)^i [\beta\xi\pi_{t+i+1} + (1 - \beta\xi)(mc_{t+i} + ms_{t+i})]
\end{aligned}
\tag{33}
$$

Then, in a symmetric equilibrium, we have

$$\pi_t = \frac{(1 - \xi)}{\xi} \left( \mathbb{E}_t \sum_{i=0}^{\infty} (\beta\xi)^i [\beta\xi\pi_{t+i+1} + (1 - \beta\xi)(mc_{t+i} + ms_{t+i})] \right) \tag{34}$$

where $\mathbb{E}_t[\pi_{t+i+1}]$ and $\mathbb{E}_t[mc_{t+i} + ms_{t+i}]$ are expectations of aggregate inflation and real marginal costs, both variables exogenous to individual price setters. However, if price setters know they are identical, they know the aggregate price level over non-optimising and optimising firms

$$p_t(m) = \xi p_{t-1} + (1 - \xi)p_t^o(m) \tag{35}$$

to obtain in a symmetric equilibrium

$$p_t^o(m) - p_t = p_t^o - p_t = \frac{\xi}{(1 - \xi)}(p_t - p_{t-1}) = \frac{\xi}{(1 - \xi)}\pi_t$$

Then, substituting back into (33), we arrive at

$$\pi_t = \frac{(1 - \xi)(1 - \beta\xi)}{\xi}\mathbb{E}_t^* \sum_{i=0}^{\infty} \beta^i (mc_{t+i} + ms_{t+i}) \tag{36}$$

which omits learning about aggregate inflation. Equation (36) is the familiar linearised Phillips curve. Under RE, (34) and (36) are equivalent. (Putting $mc_t = w_t - a_t + h_t = (1 + \phi)h_t = \frac{(1+\phi)(y_t - a_t)}{\alpha}$, (36) in recursive form gives (2) with $\lambda = \frac{(1-\xi)(1-\beta\xi)(1+\phi)}{\alpha\xi}$ and $u_{2,t} = \lambda ms_t$). The form of the Phillips curve (36), which is equivalent to (2), is often used in the behavioural NK literature (see, for example, Reference [4]), but as we have shown, this assumes that firms know they are identical. In our BR model, we use (29) and (34), which do not make this assumption.

## 4. AU Learning and Market-Consistent Information

With anticipated utility (AU) learning, our learning model is one where agents make fully optimal decisions, given their individual specification of beliefs, but have no macroeconomic model to form expectations of aggregate variables. We draw a clear distinction between aggregate and internal quantities so that identical agents in our model are not aware of this equilibrium property (nor any others).

To close the model, we need to specify the manner in which households and firms form their expectations. To do so, we assume that variables which are local to the agents, in a geographical sense, are observable within the period, whereas variables that are strictly macroeconomic are only observable with a lag. This categorisation regarding information about the current state of the economy follows Reference [29], which distinguishes between the local information that agents acquire directly through their interactions in markets and statistics that are collected and summarised, usually by governments, and are made available to the wider public. (This paper actually focuses on a third category, information provided by the news media, and allows for imperfect information in the form of noisy signals, issues which go beyond the scope of our paper.) The policy rate is announced by the central bank; thus, it is observed without a lag, and it is common knowledge. Given this, we assume an adaptive expectations forecasting rule given below by (38) and (39) about variables external to agents' decisions. Let $x_t = r_t, r_{n,t}, \pi_t, w_t, \gamma_t$, then household expectations are given by

$$\mathbb{E}_t^* x_{t+i} = \mathbb{E}_t^* x_{t+1}; \quad i \geq 1 \tag{37}$$

Expressing $\mathbb{E}_t \omega_{1,t+1}$ and $\mathbb{E}_t \omega_{2,t+1}$ in (29) as forward-looking summations and using (37), we arrive at the individual learning consumption equation

$$
\begin{aligned}
\alpha_1 c_t &= \alpha_2 w_t + \alpha_3 (\omega_{2,t} + r_t) + \alpha_4 \omega_{1,t} \\
\omega_{1,t} &= \frac{1}{1-\beta} \left[ \alpha_5 \mathbb{E}_t^* w_{t+1} - \alpha_6 (\beta \mathbb{E}_t^* r_{n,t+1} - \mathbb{E}_{h,t}^* \pi_{t+1}) \right] - \alpha_6 r_{n,t} \\
\omega_{2,t} &= (1-\beta)(\gamma_t - g_t) - r_t + \frac{\beta}{1-\beta} \left( (1-\beta)(\mathbb{E}_t^* \gamma_{t+1} - \mathbb{E}_t^* g_{t+1}) - \mathbb{E}_t^* r_{t+1} \right)
\end{aligned}
$$

which is now expressed in terms of one-step ahead forecasts by

$$\mathbb{E}_t^* x_{t+1} = \mathbb{E}_t^* x_t + \lambda_x (x_{t-j} - \mathbb{E}_t^* x_t); \quad x = w, r_n, \pi, \gamma; \quad j = 0, 1 \tag{38}$$

Households make inter-temporal decisions for their consumption and hours supplied given adaptive expectations of the wage rate, the nominal interest rate, inflation and profits. These macro-variables may in principle be observed with or without a one-period lag ($j = 1, 0$), but as stated earlier, we assume $j = 0$ for market-specific variables $w_t$, $\gamma_t$, and $j = 1$ for aggregate inflation $\pi_t$. However, we assume that the current nominal interest rate, $r_{n,t}$, is announced and therefore is observed without a lag.

We distinguish household and firm expectations $\mathbb{E}_{h,t}^* \pi_{t+1}$, $\mathbb{E}_{f,t}^* \pi_{t+1}$. Then, for retail firm $m$

$$
\begin{aligned}
\mathbb{E}_t^* \pi_{t+i+1} &= \mathbb{E}_t^* \pi_{t+1}; \quad i \geq 0 \\
\mathbb{E}_t^* (mc_{t+i} + ms_{t+i}) &= \mathbb{E}_t^* (mc_{t+1} + ms_{t+1}); \quad i \geq 1 \\
p_t^o(m) - p_t &= \frac{\beta \xi}{1-\beta} \mathbb{E}_{f,t}^* \pi_{t+1} + (1 - \beta \xi)(mc_t + ms_t) + \frac{\beta}{1-\beta} \mathbb{E}_t^* (mc_{t+1} + ms_{t+1})
\end{aligned}
$$

where one-step ahead forecasts are given by the adaptive expectations rule

$$\mathbb{E}_t^* x_{t+1} = \mathbb{E}_t^* x_t + \lambda_x (x_{t-j} - \mathbb{E}_t^* x_t); \quad x = \pi, (mc + ms); \quad j = 0, 1 \tag{39}$$

Retail firms make inter-temporal decisions for their price and output given adaptive expectations of the aggregate inflation rate and their post-shock real marginal shock wage

rate. As before, these variables may be observed with or without a one-period lag ($j = 1, 0$), but for aggregate inflation, we assume $j = 1$ as for households, but $j = 0$ for the market-specific variable $mc_t$. Note that we can in principle distinguish between households' and firms' expectations of inflation.

## 5. Heterogeneous Expectations across Agents

Now we come to the full Brock–Hommes NK model but with BR-AU rather than EL boundedly rational agents. We argue that our benchmark models, namely, an agent-level learning behavioural NK model with infinite horizon learners (AU) who use the standard Brock–Hommes forecast heuristics to form expectations, and a composite version with fixed proportions of agents forming both RE and AU in a NK setting, are selected because we want to compare the equilibrium features and empirical performance of these assumptions in an informational, consistent environment. We assume that all RE agents know the composite model, and moreover, we impose informational inconsistency by assuming that they have the same imperfect information set as the BR-AU agents. The latter do not know the model, but they make individually optimal decisions given individual observations of the states and belief formations. The composite RE-BR model then has an equilibrium (in non-linear form)

$$
\begin{aligned}
H_t^d &= n_{h,t}(H_t^s)^{RE} + (1 - n_{h,t})(H_t^s)^{BR} \\
C_t &= n_{h,t}(C_t)^{RE} + (1 - n_{h,t})(C_t)^{BR} = Y_t - G_t \\
\frac{P_t^o}{P_t} &= n_{f,t}\left(\frac{P_t^o}{P_t}\right)^{RE} + (1 - n_{f,t})\left(\frac{P_t^o}{P_t}\right)^{BR}
\end{aligned}
$$

Zero net wealth in aggregate implies that $n_{h,t}B_t^{RE} = -(1 - n_{h,t})B_t^{BR}$.

We first consider the properties of the model with fixed exogenous proportions of RE and BR agents. Then, in Section 5.2, we allow these proportions to be determined endogenously.

### 5.1. Exogenous Proportions of RE and BR Agents

For our model of BR with AU, Figure 1 plots the impulse response functions (IRFs) with standard parameters for the rule for a shock to monetary policy under fast and slow learning. Figures A3 and A4 in Appendix F show IRFs for the technology and mark-up shocks. Not surprisingly, fast learning sees an IRF converge faster to the RE case, but in either case, BR introduces *more persistence* compared with RE. This suggests that this feature should lead to a better fit of the data without relying on other persistence mechanisms (shocks, habit or price indexing). The stability properties of the model are examined in the WP version of the paper and Appendix A.

### 5.2. Endogenous Proportions of RE and BR Agents with Reinforcement Learning

Proportions of rational households ($n_{h,t}$) and firms ($n_{f,t}$) are given by (8)

$$
n_{j,t} = \frac{\exp(-\gamma \Phi_{j,t}^{RE})}{\exp(-\gamma \Phi_{j,t})^{RE} + \exp(-\gamma \Phi_{j,t}^{BR})}; \quad j = h, f
$$

where fitness for households and firms $j = h, f$ is given by

$$
\begin{aligned}
\Phi_{j,t}^{RE} &= \mu_j^{RE}\Phi_{j,t-1}^{RE} + (1 - \mu_j^{RE})\left(\text{weighted sum of forecast errors} + C_j\right) \\
\Phi_{j,t}^{BR} &= \mu_j^{BR}\Phi_{j,t-1}^{BR} + (1 - \mu_j^{BR})\left(\text{weighted sum of forecast errors}\right)
\end{aligned}
$$

Table 1 provides a third-order perturbation solution of the non-linear NK RE-BR model. We use the Bayesian estimation of the model in Reference [30] where the model is linearised and the proportions $n_{h,t}$ and $n_{f,t}$ are fixed. Non-linear estimation would be

required to pin down the parameters $n_h$, $n_f$ in the steady state in the BR scenarios and $\mu_h^{RE,BR}$, $\mu_f^{RE,BR}$ and $\gamma$ in the reinforcement learning process, which goes beyond the scope of this paper. Thus, here we impose them as reported in the table ($n_{h,t} = n_{f,t} = 0.1$). We also scale the estimated standard deviations of the shocks using a parameter $\sigma = 1, 2$. For the robustness of our results, we perform additional simulations, for different choice of the memory parameters, and present the results with $\mu_h^{RE} = \mu_h^{BR} = \mu_f^{RE} = \mu_f^{BR} = 0.5$ and $= 0.75$ in Appendix G. The robustness exercise assumes instead that agents have some memory of past observations.

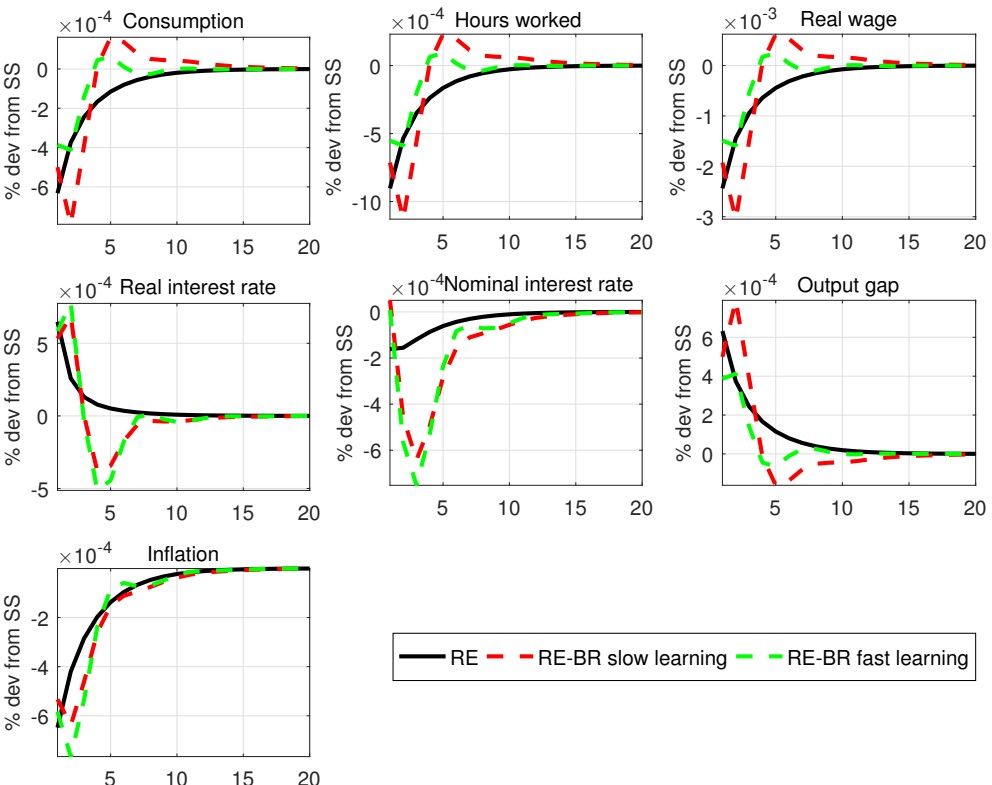

**Figure 1.** RE versus RE-BR composite expectations with $n_h = n_f = 0.5$, $\lambda_x = 0.25, 1.0$; Taylor rule with $\rho_r = 0.7$, $\theta_\pi = 1.5$ and $\theta_y = 0.3$, $\theta_{dy} = 0$; monetary policy shock.

The main results from these simulations are as follows. First, reinforcement learning introduces *high kurtosis and skewness* in macroeconomic variables, the absence of kurtosis in the standard NK model, often highlighted in the literature (see, for example, Reference [3]), is in part simply the consequence of linearisation, and non-normality is a feature of higher order approximations. Second, reinforcement learning with stronger switching processes (i.e., $\gamma = 100, 1000$) coupled with higher volatility of exogenous shocks results in the numbers of rational agents increasing from the estimated deterministic steady state value of 0.1 to 0.13 and 0.15 for households and firms, respectively, in the stochastic steady state. Third, given that BR is a welfare-reducing friction in these models, it follows that volatility can actually be welfare-increasing in our heterogeneous expectations setting. Furthermore, when we assume that agents have some memory of past observations when revising their expectations given their forecast performances, the simulated skewness and kurtosis are lower compared to the case when no memory is assumed in the learning process.

Our main results clearly suggest that, when the switching process between groups of heterogeneous agents becomes more deterministic depending on agents' willingness to learn from the past performance when predicting future outcomes, this leads to an increase in the level of rationality in the BR macroeconomy. This result is in line with the finding in Reference [3]. The cognitive effect of this selection mechanism is much stronger with the occurrence of large exogenous shocks. This group behaviour not only plays a key role in explaining the dynamic properties of the data, revaluating the importance of expectations in driving economic fluctuations in the spirit of Keynes' concept of animal spirits, but has important implications for the optimal control of policy in the spirit of the Lucas critique. Depending on intentions on the part of policymakers, the model suggests that different versions of policy can be designed and devised in a game between policymakers and the economy, with uncertainty as to which expectation formation is selected.

**Table 1.** Third-order solution of the estimated NK RE-BR model; $\mu_h^{RE} = \mu_h^{BR} = \mu_f^{RE} = \mu_f^{BR} = 0$; $\gamma = 1, 100, 1000$.

| Variable | Stochastic Mean | Standard Deviation (%) | Skewness | Kurtosis |
|---|---|---|---|---|
| $\frac{C_t}{C}$ | 0.999544 | 0.042057 | 0.323304 | 0.093034 |
| $\frac{H_t}{H}$ | 1.000273 | 0.005111 | 0.038002 | $-0.020743$ |
| $\frac{W_t}{W}$ | 0.999810 | 0.038145 | 0.318586 | 0.073488 |
| $\frac{\Pi_t}{\Pi}$ | 0.999898 | 0.004235 | $-0.045800$ | 0.030136 |
| $\frac{R_{n,t}}{R_n}$ | 0.999887 | 0.004440 | $-0.046254$ | 0.044145 |
| $\Phi_{h,t}^{RE} - C_h$ | $-0.000443$ | 0.000446 | $-2.078809$ | 6.635580 |
| $\Phi_{h,t}^{AE}$ | $-0.000526$ | 0.000516 | $-2.168947$ | 8.000489 |
| $\Phi_{f,t}^{RE} - C_f$ | $-0.000199$ | 0.000203 | $-2.279557$ | 9.082031 |
| $\Phi_{f,t}^{AE}$ | $-0.000349$ | 0.000342 | $-2.269953$ | 9.937975 |
| $n_{h,t}$ $(\gamma = 1; \sigma = 1)$ | 0.100008 | 0.000023 | 0.857638 | 4.454288 |
| $n_{f,t}$ $(\gamma = 1; \sigma = 1)$ | 0.100014 | 0.000025 | 1.586194 | 6.015115 |
| $n_{h,t}$ $(\gamma = 100; \sigma = 1)$ | 0.100750 | 0.002297 | 0.857638 | 4.454288 |
| $n_{f,t}$ $(\gamma = 100; \sigma = 1)$ | 0.101352 | 0.002479 | 1.586194 | 6.015115 |
| $n_{h,t}$ $(\gamma = 1000; \sigma = 1)$ | 0.107501 | 0.022973 | 0.857638 | 4.454288 |
| $n_{f,t}$ $(\gamma = 1000; \sigma = 1)$ | 0.113518 | 0.024787 | 1.586194 | 6.015115 |
| $n_{h,t}$ $(\gamma = 1000; \sigma = 2)$ | 0.130007 | 0.093482 | 0.888592 | 4.857691 |
| $n_{f,t}$ $(\gamma = 1000; \sigma = 2)$ | 0.154182 | 0.100265 | 1.683430 | 6.867599 |

### 5.3. The Possibility of Bifurcation and Chaotic Dynamics

Non-linear models in general open up the possibility that, for certain parameter values or initial conditions, they may exhibit chaotic dynamics. How are the obtained results related to such dynamics? This possibility is examined using the model of this paper in Reference [22].

The conclusions are: first, the RE determinacy condition for the linearised model in the vicinity of the deterministic steady state ensures local determinacy and stability in the model with a fixed proportion $n$ of fully rational agents. Second, if the linear form of the model starts from a position of indeterminacy, an increase in the fixed cost of being fully rational can lead to the loss of local stability via a Hopf bifurcation. This Hopf bifurcation appears to be super-critical, giving rise to stable limit cycles. As the speed at which agents learn increases, a rational route to randomness appears to follow, which

we explore with numerical methods. From a policy point of view, the main conclusion is that local indeterminacy about the steady state can be avoided by a careful choice of interest-rate rule that obeys a "Taylor condition" modified to allow for persistence. This is the case for our simulations which avoid chaotic dynamics.

## 6. Conclusions

This paper studies an NK behavioural model for which boundedly rational beliefs of economic agents are about payoff-relevant macroeconomic variables that are exogenous to their decision rules. Reinforcement learning is at the core of the heterogeneous expectations model and leads to the striking result that a high volatility of exogenous shocks, by assisting the learning process, can be welfare-increasing.

The results from our simulations have a range of practical and theoretical implications. From a practical point of view, our model provides a behavioural explanation for the important properties of the business cycle dynamics and (ir)rationality under market economy. Our findings shed more light on the underlying mechanism that guides policy choices in a society comprising policymakers and agents who form heterogeneous expectations. Regarding the theoretical implications, our results for a simple NK model suggest a new agenda for constructing empirical medium-sized NK models for agents' behaviours under imperfect information. Future work will embed the RE-BR composite model into a richer NK macroeconomic model along the lines of Reference [31], use non-linear estimation methods to identify a number of parameters involving reinforcement learning that are not identified using linear Bayesian estimation, and examine optimal monetary policy.

Another potential direction for future research is to investigate how reinforcement learning affects the possible chaotic dynamics of the model. We know that an increase in the fixed cost of being fully rational can lead to the loss of local stability. If we enter a region of local instability, but global boundedness, we see chaotic dynamics as highlighted generally in Reference [25]. In addition, from Reference [22], who plotted the simulated trajectories for various parameter values with an almost purely stochastic switching process ($\gamma = 0.1$), it is evident that, when the level of rationality varies according to reinforcement learning, it is likely that we see very different stability/determinacy properties of the model, which imply that uncertainty as to how expectations and learning are processed can lead to a policy rule that is unstable or has infinite multiple equilibria (i.e., is indeterminate).

As with any research, there are limitations in our study that should be addressed in future work. We have alluded to the wilderness of non-rational expectations posed by the sheer size of the literature on behavioural macroeconomics and the huge number of equilibria proposed. Any analysis based on only one choice of model clearly has limitations when turning to policy implications. A policy that works well for one particular choice may perform badly using a different model. One solution to this problem proposed by References [32,33] is to choose a policy to maximise weighted average inter-temporal welfare across a set of competing models and to weigh models based on relative forecasting performance. In other studies, the proportions of rational and non-rational agents are fixed; a possible avenue for future research would be to extend the analysis to time-varying endogenous proportions as in this paper.

Finally, there remains a wide range of views over the asymmetric macroeconomic effects of economic shocks (e.g., news, energy and monetary policy) as well as over the variations in these effects with respect to economic conditions and states. Different strands of literature offer different explanations on the existence of non-linearities, focusing on the sources of the shocks, econometric specifications and time-variation in impact and policy responses (see Reference [34] for a recent study that addresses the latter two aspects). We argue that the modelling approach and non-linear techniques used in our paper add an important dimension to this strand of literature by providing a variety of starting points for future work that investigates the non-linear effects of shocks that may originate from the time-varying nature of expectation formations and complex adaptive systems.

**Author Contributions:** Conceptualization, S.D., P.L., J.P. and B.Y.; methodology, S.D. and P.L.; software, S.D., P.L. and B.Y.; validation, P.L. and B.Y.; formal analysis, S.D., P.L. and B.Y.; investigation, S.D., P.L., J.P. and B.Y.; writing—original draft preparation, P.L. and J.P.; writing—review and editing, P.L. and B.Y.; project administration, P.L.; funding acquisition, P.L. and J.P. All authors have read and agreed to the published version of the manuscript.

**Funding:** This research was funded by the ESRC, grant number: ES/K005154/1.

**Data Availability Statement:** No data were created or analysed in this study.

**Conflicts of Interest:** The authors declare no conflict of interest.

## Appendix A. Stability Analysis

We have three possible models of expectations: rational (i.e., model consistent), boundedly rational with Euler learning and boundedly but with infinite-horizon learning. We denote these three cases by RE, EL and AU, respectively. In this section, we consider homogeneous expectations for which all agents (households and firms) form either RE or AU or EL expectations. In Section 5 of the main paper, we then allow for the possibility that households and firms are heterogenous across these groups (but retain intra-group homogeneity).

In the numerical results below, we fix parameters at their priors used in the Bayesian estimation apart from the adaptive learning parameter $\lambda_x$ which we set at unity. We make the following information assumptions: for observations of *aggregate* output and inflation, $j = 1$, which is assumed in the EL approach. In the AU approach, we need to model observations of market-specific variables consisting of factor prices, profits and marginal costs. These we assume can be observed without a lag, and therefore, $j = 0$. Note this only applies to the EL and AU agents, but the RE equilibrium assumes perfect information where agents observe all current values of state variables. However, for rational agents, the stability conditions considered now can be derived from a perfect foresight equilibrium and are independent of the information assumption.

Figure A1 compares the models in the $(\rho_r, \theta_\pi)$ space with $\theta_y = 0.3$ and $\theta_{dy} = 0$. Figure A2 sets $\rho_r = 1$ and compares the EL and AU models in $(\theta_y, \theta_\pi)$ space having re-parameterised the rule as $r_{n,t} = \rho_r r_{n,t-1} + \theta_\pi \pi_t + \theta_y y_t$. Note that this rule reduces to a price-level rule when $\theta_y = 0$. The differences in the sizes of the policy spaces that result in a saddle-path stable equilibrium are significant. Furthermore, a clear ranking of the sizes of these spaces emerges with $RE \supset EL \supset AU$. This means that, unless the policy rule is designed for the AU model, uncertainty as to which model of expectations is correct can lead to a rule that is unstable or has infinite multiple equilibria (i.e., is indeterminate).

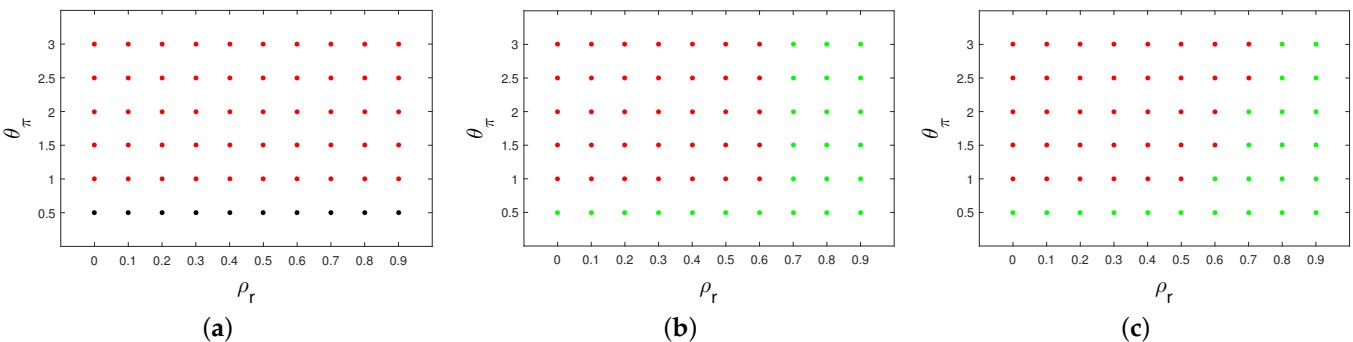

**Figure A1.** Comparison of stability properties of RE, EL and AU models in $(\rho_r, \theta_\pi)$ space; $\rho_r > 0$, $\lambda_x = 1$; red: determinacy; black: indeterminacy; green: instability. (**a**) RE: $\theta_y = 0.3$; (**b**) EL: $\theta_y = 0.3$; (**c**) AU: $\theta_y = 0.3$.

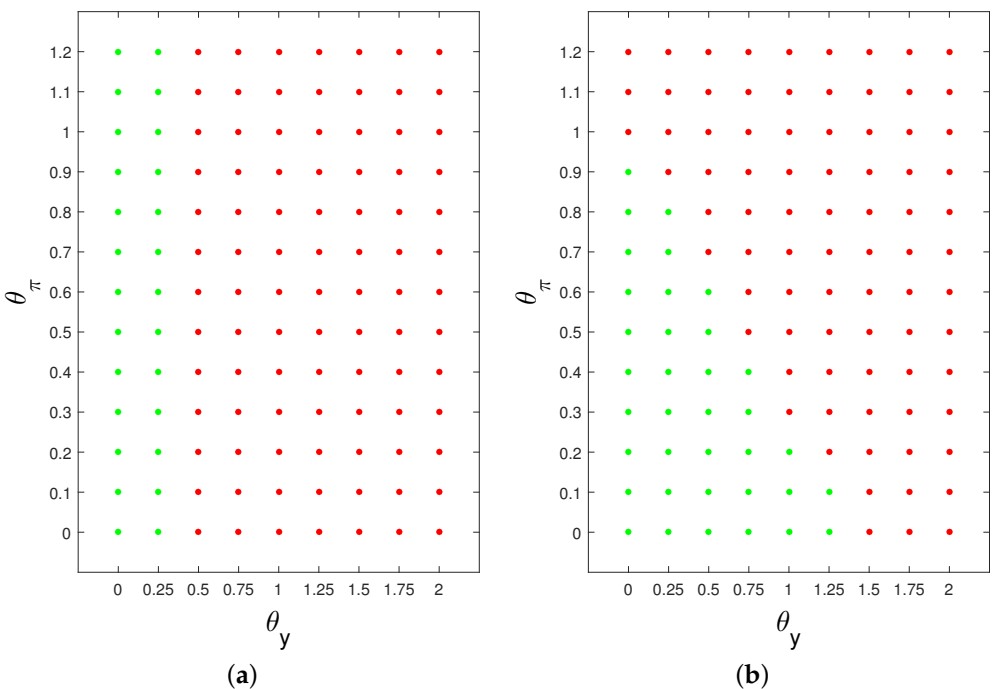

**Figure A2.** Comparison of stability properties of EL and AU models in $(\theta_y, \theta_\pi)$ space; $\rho_r = 1$, $\lambda_x = 1$; red: determinacy; black: indeterminacy; green: instability. (**a**) EL: $\rho_r = 1$; (**b**) AU: $\rho_r = 1$.

## Appendix B. Summary of Composite RE-BR Model

In stationarised form of the model for exogenous proportions $n_{h,t}$ and $n_{f,t}$, we have

**RE Households:**

$$
\begin{aligned}
U_t^{RE} &= U(C_t^{RE}, H_t^{RE}) = \log C_t^{RE} - \frac{(H_t^{RE})^{1+\phi}}{1+\phi} \\
U_{C,t}^{RE} &= \mathbb{E}_t[\beta_{g,t+1} U_{C,t+1}^{RE} R_{t+1}] \\
\beta_{g,t} &= \beta/(1+g_t) \\
g_t &= (1+g)\exp(\epsilon_{Atrend}) - 1 \\
R_t &= \frac{R_{n,t-1}}{\Pi_t} \\
U_{C,t}^{RE} &= \frac{1}{C_t^{RE}} \\
U_{H,t}^{RE} &= -(H_t^{RE})^\phi \\
-\frac{U_{H,t}^{RE}}{U_{C,t}^{RE}} &= W_t
\end{aligned}
$$

**BR Households:**

$$U_t^{BR} = U(C_t^{BR}, H_t^{BR}) = \log C_t^{BR} - \frac{(H_t^{BR})^{1+\phi}}{1+\phi}$$

$$\frac{C_t^{BR}}{(1 - \mathbb{E}_t \beta_{g,t+1})} = \frac{1}{(C_t^{BR})^{\frac{1}{\phi}}} \left( W_t^{1+\frac{1}{\phi}} + \frac{\left( \left( \frac{\mathbb{E}_t^* R_{n,t+1}}{R_{n,t}} \right) \mathbb{E}_t^* W_{t+1} \right)^{1+\frac{1}{\phi}}}{(\mathbb{E}_t \beta_{g,t+1})^{\frac{1}{\phi}} (\mathbb{E}_t^* R_{t+1}^{ex})^{1+\frac{1}{\phi}} - 1} \right)$$

$$+ \; \Gamma_t - G_t + \frac{\left( \frac{\mathbb{E}_t^* R_{n,t+1}}{R_{n,t}} \right) \mathbb{E}_t^* (\Gamma_{t+1} - G_{t+1})}{\mathbb{E}_t^* R_{t+1}^{ex} - 1}$$

$$\equiv \frac{1}{(C_t^{BR})^{\frac{1}{\phi}}} \left( W_t^{1+\frac{1}{\phi}} + \left( \frac{\mathbb{E}_t^* R_{n,t+1}}{R_{n,t}} \right)^{1+\frac{1}{\phi}} \Omega_{1,t} \right)$$

$$+ \; \Gamma_t - G_t + \left( \frac{\mathbb{E}_t^* R_{n,t+1}}{R_{n,t}} \right) \Omega_{2,t}$$

$$U_{C,t}^{BR} = \frac{1}{C_t^{BR}}$$

$$U_{H,t}^{BR} = -(H_t^{BR})^{\phi}$$

$$-\frac{U_{H,t}^{BR}}{U_{C,t}^{BR}} = W_t$$

where

$$\Omega_{1,t} = \frac{(\mathbb{E}_t^* W_{t+1})^{1+\frac{1}{\phi}}}{(\mathbb{E}_t \beta_{g,t+1})^{\frac{1}{\phi}} (\mathbb{E}_t^* R_{t+1}^{ex})^{1+\frac{1}{\phi}} - 1}$$

$$\Omega_{2,t} = \frac{\mathbb{E}_t^* (\Gamma_{t+1} - G_{t+1})}{\mathbb{E}_t^* R_{t+1}^{ex} - 1}$$

$$\mathbb{E}_t^* R_{t+1}^{ex} = \frac{\mathbb{E}_t^* R_{n,t+1}}{\mathbb{E}_{h,t}^* \Pi_{t+1}}$$

**Wholesale Firms:**

$$Y_t^W = F(A_t, H_t) = A_t H_t^{\alpha} = A_t (n_{h,t} H_t^{RE} + (1 - n_{h,t}) H_t^{BR})^{\alpha}$$

$$Y_t = \frac{Y_t^W}{\Delta_t^{\alpha}}$$

$$\frac{P_t^W}{P_t} F_{H,t} = \frac{P_t^W}{P_t} \frac{\alpha Y_t^W}{H_t} = W_t$$

$$1 = \xi \Pi_t^{\zeta - 1} + (1 - \xi) \left( n_{f,t} \left( \frac{J_t^{RE}}{JJ_t^{RE}} \right)^{1-\zeta} + (1 - n_{f,t}) \left( \frac{J_t^{BR}}{JJ_t^{BR}} \right)^{1-\zeta} \right)$$

$$\Delta_t = \xi \Pi_t^{\frac{\zeta}{\alpha}} \Delta_{t-1} + (1 - \xi) \left( n_{f,t} \left( \frac{J_t^{RE}}{JJ_t^{RE}} \right)^{-\frac{\zeta}{\alpha}} + (1 - n_{f,t}) \left( \frac{J_t^{BR}}{JJ_t^{BR}} \right)^{-\frac{\zeta}{\alpha}} \right)$$

$$MC_t = \frac{P_t^W}{P_t} = \frac{W_t}{F_{H,t}}$$

$$\Gamma_t = Y_t - \alpha MC_t Y_t^W$$

**RE Retail Firms:**

$$JJ_t^{RE} - \xi E_t[\Pi_{t+1}^{\zeta-1} JJ_{t+1}^{RE} \beta_{g,t+1}] \;=\; Y_t\left(n_{h,t} U_{C,t}^{RE} + (1 - n_{h,t}) U_{C,t}^{BR}\right)$$

$$J_t^{RE} - \xi E_t[\Pi_{t+1}^{\zeta} J_{t+1}^{RE} \beta_{g,t+1}] \;=\; \left(\frac{1}{1 - \frac{1}{\zeta}}\right) Y_t MC_t MS_t \left(n_{h,t} U_{C,t}^{RE} + (1 - n_{h,t}) U_{C,t}^{BR}\right)$$

$$\left(\frac{P_t^0}{P_t}\right)^{RE} \;=\; \frac{J_t^{RE}}{JJ_t^{RE}}$$

**BR Retail Firms:**

$$J_t^{BR} \;=\; \left(\frac{1}{1 - \frac{1}{\zeta}}\right)(Y_t MC_t MS_t + \Omega_{3,t})$$

$$JJ_t^{BR} \;=\; Y_t + \Omega_{4,t}$$

$$\left(\frac{P_t^0}{P_t}\right)^{BR} \;=\; \frac{J_t^{BR}}{JJ_t^{BR}}$$

where

$$\Omega_{3,t} \;=\; \frac{\xi(\mathbb{E}_{f,t}^* \Pi_{t+1})^{\zeta} \mathbb{E}_t^* Y_{t+1} \mathbb{E}_t^* MC_{t+1} \mathbb{E}_t^* MS_{t+1}}{\mathbb{E}_{f,t}^* R_{t+1} - \xi(\Pi_{t+1})^{\zeta}}$$

$$\Omega_{4,t} \;=\; \frac{\xi(\mathbb{E}_{f,t}^* \Pi_{t+1})^{\zeta-1} \mathbb{E}_t^* Y_{t+1}}{\mathbb{E}_{f,t}^* R_{t+1} - \xi(\mathbb{E}_{f,t}^* \Pi_{t+1})^{\zeta-1}}$$

$$\mathbb{E}_{f,t}^* R_{t+1} \;=\; \mathbb{E}_{f,t}^*\left[\frac{R_{n,t}}{\Pi_{t+1}}\right] = \frac{R_{n,t}}{\mathbb{E}_{f,t}^* \Pi_{t+1}}$$

**One-Period Ahead Adaptive Expectations:**

$$\mathbb{E}_t^*[\beta_{g,t+1}] \;=\; \mathbb{E}_{t-1}^*[\beta_g, t] + \lambda_{1,\beta_g}\left(\beta_{g,t-1} - \mathbb{E}_{t-1}^*[\beta_g, t]\right) + \lambda_{2,\beta_g}\left(\beta_{g,t-1} - \beta_{g,t-2}\right); \; \lambda_{i,\beta_g} \in [0,1]$$

$$\mathbb{E}_t^*[G_{t+1}] \;=\; \mathbb{E}_{t-1}^*[G_t] + \lambda_{1,G}\left(G_t - \mathbb{E}_{t-1}^*[G_t]\right) + \lambda_{2,G}(G_t - G_{t-1}); \; \lambda_{i,G} \in [0,1]$$

$$\mathbb{E}_t^*[W_{t+1}] \;=\; \mathbb{E}_{t-1}^*[W_t] + \lambda_W\left(W_t - \mathbb{E}_{t-1}^*[W_t]\right) + \lambda_{2,W}(W_t - W_{t-1}); \; \lambda_{i,W} \in [0,1]$$

$$\mathbb{E}_t^*[\Gamma_{t+1}] \;=\; \mathbb{E}_{t-1}^*[\Gamma_t] + \lambda_{1,\Gamma}\left(\Gamma_t - \mathbb{E}_{t-1}^*[\Gamma_t]\right) + \lambda_{2,\Gamma}(\Gamma_t - \Gamma_{t-1}); \; \lambda_{i,\Gamma} \in [0,1]$$

$$\mathbb{E}_t^*[R_{n,t+1}] \;=\; \mathbb{E}_{t-1}^*[R_{n,t}] + \lambda_{1,R_n}\left(R_{n,t} - \mathbb{E}_{t-1}^*[R_{n,t}]\right) + \lambda_{2,R_n}(R_{n,t} - R_{n,t-1}); \; \lambda_{i,R_n} \in [0,1] \text{ (households)}$$

$$\mathbb{E}_{h,t}^*[\Pi_{t+1}] \;=\; \mathbb{E}_{t-1}^*[\Pi_t] + \lambda_{1h,\Pi}\left(\Pi_{t-1} - \mathbb{E}_{t-1}^*[\Pi_t]\right) + \lambda_{2h,\Pi}(\Pi_{t-1} - \Pi_{t-2}); \; \lambda_{ih,\Pi} \in [0,1] \text{ (households)}$$

$$\mathbb{E}_{f,t}^*[\Pi_{t+1}] \;=\; \mathbb{E}_{t-1}^*[\Pi_t] + \lambda_{1f,\Pi}\left(\Pi_{t-1} - \mathbb{E}_{t-1}^*[\Pi_t]\right) + \lambda_{2h,\Pi}(\Pi_{t-1} - \Pi_{t-2}); \; \lambda_{if,\Pi} \in [0,1] \text{ (firms)}$$

$$\mathbb{E}_t^*[Y_{t+1}] \;=\; \mathbb{E}_{t-1}^*[Y_t] + \lambda_{1,Y}\left(Y_{t-1} - \mathbb{E}_{t-1}^*[Y_t]\right) + \lambda_{2,Y}(Y_{t-1} - Y_{t-2}); \; \lambda_{i,Y} \in [0,1]$$

$$\mathbb{E}_t^*[\tilde{MC}_{t+1}] \;=\; \mathbb{E}_{t-1}^*[\tilde{MC}_t] + \lambda_{1,MC}\left(\tilde{MC}_t - \mathbb{E}_{t-1}^*[\tilde{MC}_t]\right) + \lambda_{2,MC}(\tilde{MC}_t - \tilde{MC}_{t-1}); \; \lambda_{i,MC} \in [0,1]$$

where $\tilde{MC}_t \equiv MC_t MS_t$. Note that we have used the first-order approximation $\log \frac{X_t}{X} \approx \frac{X_t - X}{X}$.

**Wealth Distribution:**

First, define bond holdings of BR households by

$$B_t^{BR} = R_t B_{t-1}^{BR} + W_t H_t^{BR} + \Gamma_t - C_t^{BR} - T_t - \frac{\omega}{2}(B_{t-1}^{BR} - B)^2$$

having introduced a portfolio cost adjustment with a small $\omega$. Then, replace $C_t^{BR}$ and the Euler equation above with

$$
\begin{aligned}
\frac{C_t^{BR} - B_t^{BR}}{(1 - \mathbb{E}_t^* \beta_{g,t+1})} &= \frac{1}{(C_t^{BR})^{\frac{1}{\phi}}} \left( W_t^{1+\frac{1}{\phi}} + \frac{\left( \left( \frac{\mathbb{E}_t^* R_{n,t+1}}{R_{n,t}} \right) \mathbb{E}_t^* W_{t+1} \right)^{1+\frac{1}{\phi}}}{(\mathbb{E}_t^* \beta_{g,t+1})^{\frac{1}{\phi}} (\mathbb{E}_t^* R_{t+1}^{ex})^{1+\frac{1}{\phi}} - 1} \right) + \Gamma_t - G_t \\
&\quad + \frac{\left( \frac{\mathbb{E}_t^* R_{n,t+1}}{R_{n,t}} \right) \mathbb{E}_t^* (\Gamma_{t+1} - G_{t+1})}{\mathbb{E}_t^* R_{t+1}^{ex} - 1} \\
&\equiv \frac{1}{(C_t^{BR})^{\frac{1}{\phi}}} \left( W_t^{1+\frac{1}{\phi}} + \left( \frac{\mathbb{E}_t^* R_{n,t+1}}{R_{n,t}} \right)^{1+\frac{1}{\phi}} \Omega_{1,t} \right) \\
&\quad + \Gamma_t - G_t + \left( \frac{\mathbb{E}_t^* R_{n,t+1}}{R_{n,t}} \right) \Omega_{2,t} \\
U_{C,t}^{RE} &= \mathbb{E}_t \left[ \beta_{g,t+1} U_{C,t+1}^{RE} (R_{t+1} - \omega(B_t^{RE} - B)) \right]
\end{aligned}
$$

where zero net wealth implies $n_{h,t} B_t^{RE} = -(1 - n_{h,t}) B_t^{BR}$.

**Closure of Model:**

$$
\begin{aligned}
Y_t &= n_{h,t} C_t^{RE} + (1 - n_{h,t}) C_t^{BR} + G_t \\
G_t &= T_t \\
\log \left( \frac{R_{n,t}}{R_n} \right) &= \rho_r \log \left( \frac{R_{n,t-1}}{R_n} \right) + (1 - \rho_r) \left( \theta_\pi \log \left( \frac{\Pi_t}{\Pi_{targ,t}} \right) \right. \\
&\quad + \left. \theta_y \log \left( \frac{Y_t}{Y} \right) + \theta_{dy} \log \left( \frac{Y_t}{Y_{t-1}} \right) \right) + \epsilon_{MP,t} \\
\log A_t - \log A &= \rho_A (\log A_{t-1} - \log A) + \epsilon_{A,t} \\
\log G_t - \log G &= \rho_G (\log G_{t-1} - \log G) + \epsilon_{G,t} \\
\log MS_t - \log MS &= \rho_{MS} (\log MS_{t-1} - \log MS) + \epsilon_{MS,t} \\
\log \Pi_{targ,t} - \log \Pi &= \rho_\pi (\log \Pi_{targ,t-1} - \log \Pi) + \epsilon_{\pi,t}
\end{aligned}
$$

**Endogenous Proportions of RE and BR Agents:**

The payoff for households and firms is expressed in terms of a discounted sum of past weighted forecast errors, $\Phi_{h,t}$ say, starting at $t = 0$ for rational and non-rational households, respectively,

$$
\begin{aligned}
\Phi_{h,t}^{RE} &= \mu_h^{RE} \Phi_{h,t-1}^{RE} - (1 - \mu_h^{RE}) \left( w_{\beta_g} (\beta_{g,t} - E_{h,t-1} \beta_{g,t})/\beta_g)^2 + w_G ((G_t - E_{h,t-1} G_t)/G)^2 \right. \\
&\quad + w_W ((W_t - E_{h,t-1} W_t)/W)^2 + w_{h,\Pi} ((\Pi_t - E_{h,t-1}\Pi)/\Pi)^2 \\
&\quad + \left. w_\Gamma ((\Gamma_t - E_{h,t-1}\Gamma_t)/\Gamma)^2 + w_R ((R_{n,t} - E_{t-1} R_{n,t})/R_n)^2 + C_h \right) \\
\Phi_{h,t}^{BR} &= \mu_h^{BR} \Phi_{h,t-1}^{BR} - (1 - \mu_h^{BR}) \left( w_{\beta_g} (\beta_{g,t} - E_{h,t-1}^* \beta_{g,t})/\beta_g)^2 + w_G ((G_t - E_{h,t-1}^* G_t)/G)^2 \right. \\
&\quad + w_W ((W_t - E_{h,t-1}^* W_t)/W)^2 + w_{h,\Pi} ((\Pi_t - E_{h,t-1}^* \Pi)/\Pi)^2 + w_\Gamma ((\Gamma_t - E_{h,t-1}^* \Gamma_t)/\Gamma)^2 \\
&\quad + \left. w_R ((R_{n,t} - E_{t-1} R_{n,t})/R_n)^2 \right)
\end{aligned}
$$

The parameter $C_h$ is a fixed cost of being rational for households. For firms, this becomes

$$
\begin{aligned}
\Phi_{f,t}^{RE} &= \mu_f^{RE}\Phi_{f,t-1}^{RE} - (1-\mu_f^{RE})\Big(w_Y((Y_t - E_{f,t-1}Y_t)/Y)^2 + w_{f,\Pi}((\Pi_t - E_{f,t-1}\Pi)/\Pi)^2 \\
&\quad + w_{MC}((\tilde{M}C_t - E_{f,t-1}\tilde{M}C_t)/MC)^2 + C_f\Big) \\
\Phi_{f,t}^{BR} &= \mu_f^{BR}\Phi_{f,t-1}^{BR} - (1-\mu_f^{BR})\Big(w_Y((Y_t - E_{f,t-1}^*Y_t)/Y)^2 + w_{f,\Pi}((\Pi_t - E_{f,t-1}^*\Pi)/\Pi)^2 \\
&\quad + w_{MC}((\tilde{M}C_t - E_{f,t-1}^*\tilde{M}C_t)/MC)^2\Big)
\end{aligned}
$$

where parameter $C_f$ is a fixed cost of being rational for firms, and we allow for the possibility that $C_h \neq C_f$. Then, the proportions of rational households and firms is given by

$$
\begin{aligned}
n_{h,t} &= \frac{\exp(\gamma\Phi_{h,t}^{RE})}{\exp(\gamma\Phi_{h,t})^{RE} + \exp(\gamma\Phi_{h,t}^{BR})} = \frac{\exp(\gamma(\Phi_{h,t}^{RE} - \Phi_{h,t}^{BR}))}{\exp(\gamma(\Phi_{h,t}^{RE} - \Phi_{h,t}^{BR})) + 1} \\
n_{f,t} &= \frac{\exp(\gamma\Phi_{f,t}^{RE})}{\exp(\gamma\Phi_{f,t})^{RE} + \exp(\gamma\Phi_{f,t}^{BR})} = \frac{\exp(\gamma(\Phi_{f,t}^{RE} - \Phi_{f,t}^{BR}))}{\exp(\gamma(\Phi_{f,t}^{RE} - \Phi_{f,t}^{BR})) + 1}
\end{aligned}
$$

Thus, the proportion of rational agents in the steady state is given by

$$
\begin{aligned}
n_h &= \frac{\exp(-\gamma C_h)}{\exp(-\gamma C_h) + 1} \\
n_f &= \frac{\exp(-\gamma C_f)}{\exp(-\gamma C_f) + 1}
\end{aligned}
$$

which is pinned down by the cost parameters $(C_h, C_f)$ (which can be positive or negative).

**Welfare and Consumption Equivalence:**

$$
\begin{aligned}
U_t &= \log((n_{h,t}C_t^{RE} + (1-n_{h,t})C_t^{BR}) - \frac{(n_{h,t}H_t^{RE} + (1-n_{h,t}H_t)^{BR})^{1+\phi}}{1+\phi} \\
wel_t &= (1-\beta_{g,t})U_t + \mathbb{E}_t[\beta_{g,t+1}wel_{t+1}] \\
wel_t^{RE} &= (1-\beta_{g,t})U_t^{RE} + \mathbb{E}_t[\beta_{g,t+1}wel_{t+1}^{RE}] \\
wel_t^{BR} &= (1-\beta_{g,t})U_t^{BR} + \mathbb{E}_t[\beta_{g,t+1}wel_{t+1}^{BR}] \\
CE_t &= \log(1.01C_t) - \log(C_t)
\end{aligned}
$$

**Appendix C. Balanced Growth Steady State**

In recursive form, the zero-growth zero-inflation ($\Pi = 1$) steady state can be written as

$$
\begin{aligned}
R &= \frac{1}{\beta} \\
\Lambda &= \beta \\
MC = \frac{P^W}{P} &= 1 - \frac{1}{\zeta} \\
\frac{C}{Y} &= 1 - g_y \\
H &= \frac{\alpha \Delta^\alpha MC}{\kappa (1 - g_y)} \\
Y^W &= (AH)^\alpha \\
Y &= \frac{Y^W}{\Delta^\alpha} \\
W &= \alpha \frac{P^W}{P} \frac{Y^W}{H} \\
J &= \frac{Y MC U_C}{(1 - \frac{1}{\zeta})(1 - \xi \beta \Pi^\zeta)} \\
JJ &= \frac{Y U_C}{(1 - \xi \beta \Pi^{\zeta-1})} \\
\text{Hence, with } \Pi = 1, J &= JJ \\
\Delta &= 1 \\
\Gamma &= Y - \alpha MC Y^W
\end{aligned}
$$

For a particular steady state, the inflation rate $\Pi > 1$, and the NK features of the steady state become

$$
\begin{aligned}
\frac{J}{JJ} &= \left( \frac{1 - \xi \Pi^{\zeta-1}}{1 - \xi} \right)^{\frac{1}{1-\zeta}} \\
MC = \frac{P^W}{P} &= \left( 1 - \frac{1}{\zeta} \right) \frac{J(1 - \beta \xi \Pi^\zeta)}{JJ(1 - \beta \xi \Pi^{\zeta-1})} \\
\Delta &= \frac{(1 - \xi)^\alpha \left( \frac{J}{JJ} \right)^{-\zeta}}{1 - \xi \Pi^\zeta}
\end{aligned}
$$

then, $P^W Y^W / PY = MC\Delta$.

We can now easily set up the model with a balanced exogenous-growth steady state. Now the process for $A_t$ is replaced with

$$
\begin{aligned}
A_t &= \bar{A}_t A_t^c \\
\bar{A}_t &= (1 + g)\bar{A}_{t-1} \exp(\epsilon_{A,t}) \\
\log A_t^c - \log A^c &= \rho_A (\log A_{t-1}^c - \log A^c) + \epsilon_{A,t}
\end{aligned}
$$

where $A_t$ is a labour-augmenting technical progress parameter which we decompose into a cyclical component, $A_t^c$, modelled as a temporary AR(1) process and a stochastic trend, whose log is a random walk with drift, $\bar{A}_t$. Thus, the balanced growth deterministic steady state path is driven by labour-augmenting technical change growing at a net rate $g$. If we put $g = \epsilon_{trend,t} = 0$ and $\bar{A}_t = 1$, we arrive at our previous formulation with $A_t^c = A_t$.

Now stationarise the variables by defining cyclical and stationary components

$$(Y_t^W)^c \equiv \frac{Y_t^W}{\bar{A}_t} = A_t^c H_t^\alpha$$

$$C_t^c \equiv \frac{C_t}{\bar{A}_t}$$

$$W_t^c \equiv \frac{W_t}{\bar{A}_t}$$

$$U_t^c \equiv \log C_t^c - \kappa \frac{H_t^{1+\phi}}{1+\phi}$$

$$U_{C,t}^c \equiv \frac{1}{C_t^c}$$

$$\Lambda_{t,t+1} = \beta \frac{U_{C,t+1}}{U_{C,t}} = \beta_{g,t+1} \frac{U_{C,t+1}^c}{U_{C,t}^c}$$

for all non-stationary variables where

$$g_t \equiv \frac{(\bar{A}_t - \bar{A}_{t-1})}{\bar{A}_t} = (1+g)\exp(\epsilon_{A,t}) - 1$$

$$\beta_{g,t} \equiv \beta(1+g_t)$$

is the stochastic steady state growth rate; then, the stationarised Euler equation and the Calvo pricing become

$$E_t[\Lambda_{t,t+1} R_{t+1}] = E_t\left[\beta_{g,t+1} \frac{U_{C,t+1}^c}{U_{C,t}^c} R_{t+1}\right] = 1$$

and

$$\widehat{JJ}_t^c - \xi E_t[\Pi_{t+1}^{\zeta-1} \widehat{JJ}_{t+1}^c \Lambda_{t,t+1}] = Y_t^c$$

$$\widehat{J}_t^c - \xi E_t[\Pi_{t+1}^{\zeta} \widehat{J}_{t+1}^c \Lambda_{t,t+1}] = Y_t^c MC_t MS_t$$

or equivalently

$$\widehat{JJ}_t^c - \xi E_t[\Pi_{t+1}^{\zeta-1} \widehat{JJ}_{t+1}^c \beta_{g,t+1}] = Y_t^c U_t^c$$

$$\widehat{J}_t^c - \xi E_t[\Pi_{t+1}^{\zeta} \widehat{J}_{t+1}^c \beta_{g,t+1}] = Y_t^c U_t^c MC_t MS_t$$

The steady state for the rest of the system is the same as the zero-growth one except for the following relationships:

$$R = \frac{1}{\beta_g} = \frac{R_n}{\Pi}$$

where $R$ and $R_n$ are the real and nominal steady state interest rates, and $\Pi$ is inflation.

## Appendix D. Lemma

In the first-order conditions for Calvo contracts and expressions for value functions, we are confronted with expected discounted sums of the general form

$$\Omega_t = \mathbb{E}_t\left[\sum_{k=0}^{\infty} \beta^k X_{t,t+k} Y_{t+k}\right]$$

where $X_{t,t+k}$ has the property $X_{t,t+k} = X_{t,t+1}X_{t+1,t+k}$ and $X_{t,t} = 1$ (for example an inflation, interest or discount rate over the interval $[t, t+k]$).

**Lemma A1.** $\Omega_t$ *can be expressed as*

$$\Omega_t = Y_t + \beta\mathbb{E}_t[X_{t,t+1}\Omega_{t+1}]$$

**Proof.**

$$
\begin{aligned}
\Omega_t &= X_{t,t}Y_t + \mathbb{E}_t\left[\sum_{k=1}^{\infty}\beta^k X_{t,t+k}Y_{t+k}\right] \\
&= Y_t + \mathbb{E}_t\left[\sum_{k'=0}^{\infty}\beta^{k'+1}X_{t,t+k'+1}Y_{t+k'+1}\right] \\
&= Y_t + \beta\mathbb{E}_t\left[\sum_{k'=0}^{\infty}\beta^{k'}X_{t,t+1}X_{t+1,t+k'+1}Y_{t+k'+1}\right] \\
&= Y_t + \beta\mathbb{E}_t[X_{t,t+1}\Omega_{t+1}]
\end{aligned}
$$

□

**Appendix E**

**Proof of Equation (28).** In the next period, $\xi$ of these firms will keep their old prices, and $(1-\xi)$ will change their prices to $P_{t+1}^{O}$. By the law of large numbers, we assume that the distribution of prices among those firms that do not change their prices is the same as the overall distribution in period $t$. It follows that we may write

$$
\begin{aligned}
\Delta_{t+1} &= \xi\sum_{j_{no\ change}}\left(\frac{P_t(j)}{P_{t+1}}\right)^{-\zeta} + (1-\xi)\left(\frac{J_{t+1}}{JJ_{t+1}}\right)^{-\zeta} \\
&= \xi\left(\frac{P_t}{P_{t+1}}\right)^{-\zeta}\sum_{j_{no\ change}}\left(\frac{P_t(j)}{P_t}\right)^{-\zeta} + (1-\xi)\left(\frac{J_{t+1}}{JJ_{t+1}}\right)^{-\zeta} \\
&= \xi\left(\frac{P_t}{P_{t+1}}\right)^{-\zeta}\sum_{j}\left(\frac{P_t(j)}{P_t}\right)^{-\zeta} + (1-\xi)\left(\frac{J_{t+1}}{JJ_{t+1}}\right)^{-\zeta} \\
&= \xi\Pi_{t+1}^{\zeta}\Delta_t + (1-\xi)\left(\frac{J_{t+1}}{JJ_{t+1}}\right)^{-\zeta}
\end{aligned}
$$

□

## Appendix F. Additional Simulated IRFs for RE-BR Composite Models

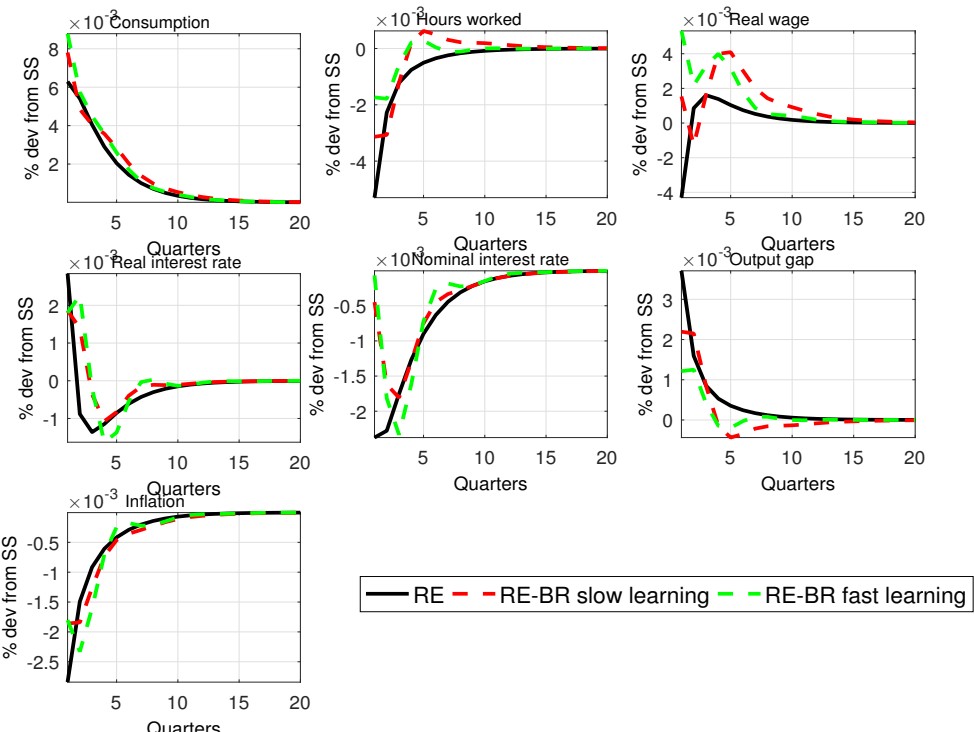

**Figure A3.** RE versus RE-BR composite expectations with $n_h = n_f = 0.5$; $\lambda_x = 0.25, 1.0$; Taylor rule with $\rho_r = 0.7$, $\theta_\pi = 1.5$ and $\theta_y = 0.3$; technology shock.

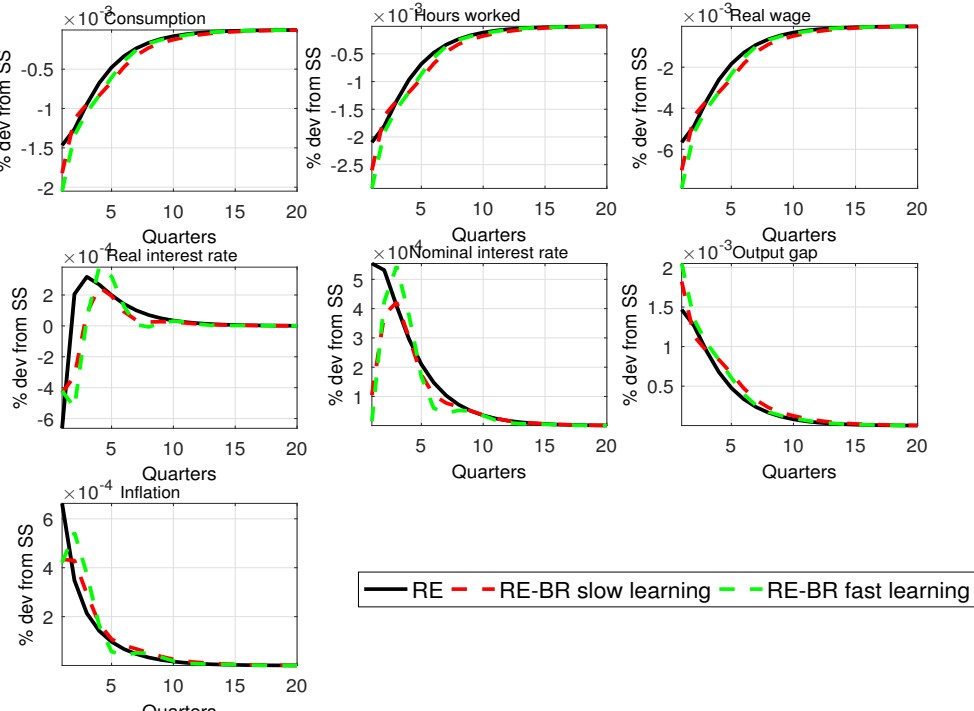

**Figure A4.** RE versus RE-BR composite expectations with $n_h = n_f = 0.5$; $\lambda_x = 0.25, 1.0$; Taylor rule with $\rho_r = 0.7$, $\theta_\pi = 1.5$ and $\theta_y = 0.3$; mark-up shock.

## Appendix G. Robustness

**Table A1.** Third-order solution of the estimated NK RE-BR model; $\mu_h^{RE} = \mu_h^{BR} = \mu_f^{RE} = \mu_f^{BR} = 0.5$; $\gamma = 1, 100, 1000$.

| Variable | Stochastic Mean | Standard Deviation (%) | Skewness | Kurtosis |
|---|---|---|---|---|
| $\frac{C_t}{C}$ | 0.999544 | 0.042057 | 0.323304 | 0.093034 |
| $\frac{H_t}{H}$ | 1.000273 | 0.005111 | 0.038002 | −0.020743 |
| $\frac{W_t}{W}$ | 0.999810 | 0.038145 | 0.318586 | 0.073488 |
| $\frac{\Pi_t}{\Pi}$ | 0.999898 | 0.004235 | −0.045800 | 0.030136 |
| $\frac{R_{n,t}}{R_n}$ | 0.999887 | 0.004440 | −0.046254 | 0.044145 |
| $\Phi_{h,t}^{RE} - C_h$ | −0.000443 | 0.000257 | −1.504159 | 3.793195 |
| $\Phi_{h,t}^{AE}$ | −0.000526 | 0.000303 | −1.592581 | 4.581412 |
| $\Phi_{f,t}^{RE} - C_f$ | −0.000199 | 0.000116 | −1.672777 | 5.558457 |
| $\Phi_{f,t}^{AE}$ | −0.000349 | 0.000226 | −1.897335 | 7.457836 |
| $n_{h,t}$ ($\gamma = 1; \sigma = 1$) | 0.100008 | 0.000013 | 0.488774 | 3.275592 |
| $n_{f,t}$ ($\gamma = 1; \sigma = 1$) | 0.100014 | 0.000016 | 1.680492 | 6.480563 |
| $n_{h,t}$ ($\gamma = 100; \sigma = 1$) | 0.100750 | 0.001295 | 0.488774 | 3.275592 |
| $n_{f,t}$ ($\gamma = 100; \sigma = 1$) | 0.101352 | 0.001568 | 1.680492 | 6.480563 |
| $n_{h,t}$ ($\gamma = 1000; \sigma = 1$) | 0.107502 | 0.012952 | 0.488774 | 3.275592 |
| $n_{f,t}$ ($\gamma = 1000; \sigma = 1$) | 0.113519 | 0.015679 | 1.680492 | 6.480563 |
| $n_{h,t}$ ($\gamma = 1000; \sigma = 2$) | 0.130010 | 0.052873 | 0.535046 | 3.638229 |
| $n_{f,t}$ ($\gamma = 1000; \sigma = 2$) | 0.154185 | 0.063624 | 1.779321 | 7.399916 |

**Table A2.** Third-order solution of the estimated NK RE-BR model; $\mu_h^{RE} = \mu_h^{BR} = \mu_f^{RE} = \mu_f^{BR} = 0.75$; $\gamma = 1, 100, 1000$.

| Variable | Stochastic Mean | Standard Deviation (%) | Skewness | Kurtosis |
|---|---|---|---|---|
| $\frac{C_t}{C}$ | 0.999544 | 0.042057 | 0.323304 | 0.093034 |
| $\frac{H_t}{H}$ | 1.000273 | 0.005111 | 0.038002 | −0.020743 |
| $\frac{W_t}{W}$ | 0.999810 | 0.038145 | 0.318586 | 0.073488 |
| $\frac{\Pi_t}{\Pi}$ | 0.999898 | 0.004235 | −0.045797 | 0.030137 |
| $\frac{R_{n,t}}{R_n}$ | 0.999887 | 0.004440 | −0.046251 | 0.044145 |
| $\Phi_{h,t}^{RE} - C_h$ | −0.000443 | 0.000170 | −0.978598 | 1.538134 |
| $\Phi_{h,t}^{AE}$ | −0.000526 | 0.000204 | −1.088202 | 2.164231 |
| $\Phi_{f,t}^{RE} - C_f$ | −0.000199 | 0.000077 | −1.063312 | 2.243911 |
| $\Phi_{f,t}^{AE}$ | −0.000349 | 0.000159 | −1.414287 | 4.290569 |
| $n_{h,t}$ ($\gamma = 1; \sigma = 1$) | 0.100008 | 0.000008 | 0.350716 | 2.281134 |
| $n_{f,t}$ ($\gamma = 1; \sigma = 1$) | 0.100014 | 0.000011 | 1.385635 | 4.243151 |
| $n_{h,t}$ ($\gamma = 100; \sigma = 1$) | 0.100750 | 0.000821 | 0.350716 | 2.281134 |
| $n_{f,t}$ ($\gamma = 100; \sigma = 1$) | 0.101352 | 0.001081 | 1.385635 | 4.243151 |
| $n_{h,t}$ ($\gamma = 1000; \sigma = 1$) | 0.107503 | 0.008211 | 0.350716 | 2.281134 |
| $n_{f,t}$ ($\gamma = 1000; \sigma = 1$) | 0.113521 | 0.010812 | 1.385635 | 4.243151 |
| $n_{h,t}$ ($\gamma = 1000; \sigma = 2$) | 0.130012 | 0.033699 | 0.406619 | 2.557592 |
| $n_{f,t}$ ($\gamma = 1000; \sigma = 2$) | 0.154191 | 0.044060 | 1.491071 | 4.993491 |

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
