# Peer review of "Reinforcement Learning in a New Keynesian Modelâ€"

_algorithms, doi:10.3390/a16060280_

Round 1
Reviewer 1 Report
The article is an interesting achievement in the field of behavioral macroeconomics. It considers a New Keynesian (NK) model that includes both fully rational (RE) agents and agents with bounded rationality (BR). The main conclusion is that reinforcement learning under increased volatility of exogenous shocks leads to an increase in the proportion of RE agents and is welfare-increasing. The article has some publication potential, but needs some revisions and clarification of a few issues.
1. The originality report shows that much of the article, 3516 words - 56%, came from a single online source: openaccess.city.ac.uk. This contradicts the authors’ claim that the text has not been previously published. I would ask for a thorough explanation of this problem.
2. There are no research hypotheses in the article. The final section shows that three issues were studied. It is therefore advisable to include three research hypotheses in the initial section of the paper. They will help organize the article and will be a suitable guideline for readers beginning to read the text.
3. The NK model under study is nonlinear, and from its equations it follows that for certain parameter values or initial conditions it may exhibit chaotic dynamics. How are the obtained results related to chaotic dynamics?
4. Have the Lyapunov exponents of the system paths or bifurcation diagrams of these exponents and the main variables of the system been calculated?
5. Can reinforcement learning reduce (or enhance) possible chaotic dynamics of the main model variables?
Author Response
We thought each reviewer would appreciate our responses to the other two so we attach a response to all three.

Reviewer 2 Report
1) In the abstract: a) Also give details about method, that is missing. b) What are the implications of the finding regarding inclined volatility leading an increase in the share of RE agents. Please state implications of this finding and highlight the contribution of the paper at the concluding sentences.
2) In the introduction, it is not clear what the aim of the paper is in the early paragraphs. Here, a throughout introduction with references to the literature are given, which is fine. However, the contribution of the paper and to what end will it serve should be further discussed. Therefore, I suggest a revision here in the introduction section to highlight the focus of the study. It is also clear that the intro is quite short and there is space to add the above-mentioned in addition to discussion with further references. Here, rationality of irrationality and why agents might rationally choose not to correct irrational behavior under market economy could be also discussed. I suggest the following references from experimental economics and psychology fields. https://journals.sagepub.com/doi/full/10.1177/00076503221101888
https://psycnet.apa.org/fulltext/2023-58265-001.html
3) Taylor rule equation is too long and it did not fit to the paper. Please correct.
4) The model has many assumptions in addition to type of rationalities. Among these, different Price level behavior is left out, interest rates are assumed as following general forms in a world that we are not living today. Zero-lower bound and price level could be added with more discussion regarding high and unexpected inflationary paths as today. Efficiency of the model could be discussed under such scenarios. In addition, model could be discussed in a discussion section under Non-Ricardian equivalence framework, Barro and Fiscal Theory of Price level (Woodford, Sims and Leeper models), where agents rationally experience wealth effects and government chooses to finance debt Bt-1 not very effectively by future primary surplusses. Further, what is the effect of changes in costs of debt instead of the amount of debt itself? With costs of debt, I don't only refer to the interest payments and interest rates, but instead, in addition to amount of interes, how frequent the payments are made under shortened maturity. Such topics are also discussed in NK literature and FTPL literature I exampled above with Woodford, Sims, Leeper's papers.
Not to effect the overall method in this paper, I suggest authors to form a discussion section under different factors. Also, models relation to recent economic situation could be discussed in this discussion section. Most basically, it could be also nice to state the implications of say a not holding transversality condition on the model and its findings. Just after line 279, a discussion section should be added, policy implications should be stated.
5) Volatility result is very striking in this paper. However, there is space to further discuss with references to existent literature with empirical findings how this finding would effect the agents and the economy as a whole.
6) Conclusion should be extended. The findings of the paper should be further highlighted here. Paper should be central. Future directions are nice however, it diminishes the overall contribution given that conclusion is too short and half of it focuses on necessity of future nonlinear methods. It is very good that authors showed they had concerns about it which are left to future studies. However, given the shortness of the conclusion, no policy implications discussions of the findings, there is space that authors should highlight the contributions of the paper.
7) Regarding nonlinearity concern: In fact, from the first equation, model started with log linearized equations. In the nonlinear section, the approach continued. In assumptions, series are assumed to follow AR(1) processes which is very general in literature and no problem about it. However, the volality conclusion of the paper is very different than the expectations. Therefore, I believe authors could discuss introductions to equations with different processes. For instance, what happens in the Taylor rule equation if the residuals follow nonlinear GARCH processes? What happens to expectations of rational and bounded rational agents under fuzzy sets or different types of volatility processes? Since the volatility finding is very central, there is space to further add certain directions regarding volatility. I open this as a debate and leave it to authors' discreation to decide on which directions and on which parts they decide to introduce such changes in the next revision. My suggestion is to extend the paper before conclusion section with such extentions.
8) Regarding my comment 7 above: I assumed that there might exist an appendix section with such extentions and I wanted to see if authors already have written a discussion regarding how the model will behave under different conditions. However, I could not see it. I also see that they noted that "A separate Online Appendix contains further details and results on the model stability and 52
the construction of the model." I don't know where, so I could not check. In fact, there is a supplementary zip file with latex codes and eps and image files. There are sooo many files in this zip file, I don't think there is an appendix that authors mentioned. I would expect it after the end of references as usual.
Author Response

(The authors gave the same response as above.)

Reviewer 3 Report
The authors have presented a New Keynesian behavioral macroeconomic model with bounded rationality and heterogeneous agents. The authors presented the formal model and the limited results of the performed simulations.
Please address the following issues:
1. Please clearly state in the introduction what are the main original contributions of this paper as compared to the state-of-the-art research.
2. The literature on the agent-based modeling and simulation and reinforcement learning is very extensive. It should be included in the related research works section, and the authors should explain how their approach relates to the state-of-the-art research on that subject.
3. Please thoroughly explain how the agent-based approach and reinforcement learning algorithms are used in the presented model. Agent-based approach is a particular branch of modeling and simulation, it has a quite established meaning, and it seems that in this paper it is used in a different context, so it should be thoroughly explained.
4. Furthermore, the use of reinforcement learning algorithms together with agent-based approach should be explained in details. Please present your approach in a form of algorithms (pseudocode).
5. The results of experiments and their interpretation are very limited - please perform more simulations, maybe for different scenarios, and present the results with thorough comments.
6. What are the advantages and disadvantages (limitations) of the proposed approach?
Author Response

(The authors gave the same response as above.)

Round 2
Reviewer 1 Report
In the new version of the paper, the authors have taken into account my advice from the previous review and tried to clarify questionable issues. Some ambiguity regarding the originality of the paper remains, as a very similar paper has already been published and it is not Deak et al. (2017):
https://openaccess.city.ac.uk/id/eprint/19621/
This item is not in the References section, although it should be. This raises a reasonable suspicion of artificially inflating the number of publications.
The authors have revised the Introduction section, in which they presented the main research questions and indicated where they contribute to the existing literature. In addition, section 5.3 has been added to the article, which addresses the chaotic dynamics of the model for certain configurations of parameter values and initial conditions. However, the problem of the relationship between reinforcement learning and the chaotic dynamics of the model remains unexplained.
To sum up, it is essential to make two corrections:
1. Include the publication https://openaccess.city.ac.uk/id/eprint/19621/ in the References section. This will avoid accusations of self-plagiarism.
2. Discuss in detail in the Conclusions section the issue of the relationship between reinforcement learning and the possible chaotic dynamics of the main variables of the model. The authors can include here their answer to my question 5 from the previous review. This issue can be presented as one of the possible directions for further research.
Author Response

(The authors gave the same response as above.)

Reviewer 2 Report
Dear Authors, I read the last version of the paper after revisions being integrated after following all reviewers' comments. Regarding my comments at my stake, this version has been improved over the previous one and the majority of my critiques are addressed in this final version which is satisfactory. My decision is positive. I noted that future direction is noted for time varying models. Also include nonlinear models, threshold models and neural networks for future work suggestion.
I also suggest including following paper to literature and/or evaluating it for future work which could be considered for calculation of sensitivity of economic variables and agents to incentives with different levels and signs nonlinearly.
Markov-switching vector autoregressive neural networks and sensitivity analysis of environment, economic growth and petrol prices. Environmental Science and Pollution Research 25 (31), 2018.
Author Response

(The authors gave the same response as above.)

Reviewer 3 Report
The authors have addressed some issues. However, there are no appendices with the complete model setup mentioned in the author's response.
Author Response

(The authors gave the same response as above.)
